# Fröhlich interaction dominated by a single phonon mode in CsPbBr$_3$

Claudiu M. Iaru [1✉], Annalisa Brodu[2], Niels J. J. van Hoof[1], Stan E. T. ter Huurne[1], Jonathan Buhot [3,4], Federico Montanarella[2], Sophia Buhbut[2], Peter C. M. Christianen [4], Daniël Vanmaekelbergh[2], Celso de Mello Donega [2], Jaime Gòmez Rivas [1], Paul M. Koenraad [1] & Andrei Yu. Silov [1✉]

The excellent optoelectronic performance of lead halide perovskites has generated great interest in their fundamental properties. The polar nature of the perovskite lattice means that electron-lattice coupling is governed by the Fröhlich interaction. Still, considerable ambiguity exists regarding the phonon modes that participate in this crucial mechanism. Here, we use multiphonon Raman scattering and THz time-domain spectroscopy to investigate Fröhlich coupling in CsPbBr$_3$. We identify a longitudinal optical phonon mode that dominates the interaction, and surmise that this mode effectively defines exciton-phonon scattering in CsPbBr$_3$, and possibly similar materials. It is additionally revealed that the observed strength of the Fröhlich interaction is significantly higher than the expected intrinsic value for CsPbBr$_3$, and is likely enhanced by carrier localization in the colloidal perovskite nanocrystals. Our experiments also unearthed a dipole-related dielectric relaxation mechanism which may impact transport properties.

[1] Department of Applied Physics and Institute for Photonic Integration, Eindhoven University of Technology, P.O. Box 513, 5600 MB Eindhoven, The Netherlands. [2] Condensed Matter and Interfaces, Debye Institute for Nanomaterials Science, Utrecht University, Princetonplein 1, 3508 TA Utrecht, The Netherlands. [3] HH Wills Laboratory, University of Bristol, Bristol BS8 1TL, UK. [4] High Field Magnet Laboratory (HFML - EMFL), Radboud University, 6525 ED Nijmegen, The Netherlands. ✉email: C.M.Iaru@tue.nl; A.Y.Silov@tue.nl

Lead halide perovskites are prime materials for optoelectronic applications, with promising results in solar cells and light-emitting devices. Despite the enthusiasm for these materials, relatively little is understood regarding the fundamental properties and mechanisms responsible for strong device performance. It has become widely accepted that the Fröhlich interaction is crucial in describing carrier behavior in the polar lattice of lead halide perovskites[1–7]. Most discussions on polaron physics in perovskites concern hybrid organic–inorganic lead iodides, from which highly efficient solar cells have been developed. These materials have a low exciton binding energy, enabling conduction with free electrons and holes that strongly polarize the lattice. However, some halide perovskite species, such as the fully inorganic $CsPbBr_3$[8], or layered two-dimensional hybrid perovskites[9] exhibit room-temperature stable excitons, which are highly desirable for light emission applications. Due to the high exciton binding energy, the Fröhlich interaction in these materials will produce excitonic polarons. Consequently, the interaction is expected to be very weak, as the phonon clouds of electron and hole polarons mutually compensate when overlapping. However, a relatively strong Fröhlich interaction occurs in $CsPbBr_3$ nanostructures[2,10], implying a reduced compensation of the electron and hole charges. It is also well-known that spectral broadening in lead halide perovskites at room temperature is dominated by the contribution of longitudinal optical (LO) phonons[2,11,12], meaning that the Fröhlich mechanism defines the color purity of light-emitting devices under ambient conditions.

We remark that the underlying reason for the excellent efficiency of perovskite-based optoelectronic devices is still under debate[13]. A number of theories exist, including the scarcity of mid-gap defect states (which would otherwise impact the emission and carrier diffusion lengths through Shockley–Read–Hall recombination), the low rate of nonradiative recombination (determined by the low energies of involved phonons), or the contribution of polaronic carrier screening. Here we address the large polaron model, which appropriately describes carrier–LO phonon interactions in $CsPbBr_3$ and related materials, as evidenced by the direct observation of the transient elastic strain fields associated with polaron formation[14]. We note, however, that anharmonic lattice vibrations may account for discrepancies between first-principles calculations within the Fröhlich model and experimentally measured carrier mobilities[15–18]. Additionally, strong anharmonicity has been shown to significantly modify the vibrational dynamics of lead halide perovskites[19]. Finally, phonon-defect scattering can lead to phonon dephasing, which may further disrupt the fully-harmonic picture of the perovskite lattice that the Fröhlich model assumes[20]. Regardless of such complications, we maintain that the Fröhlich interaction remains a driving mechanism in perovskite physics, which requires proper consideration. Indeed, it has been shown experimentally that the Fröhlich interaction shapes the room-temperature electronic band structure of $CsPbBr_3$ (ref. [21]), and is therefore essential in the description of carrier behavior in potential devices. A good starting point for probing the Fröhlich interaction is to conduct measurements at cryogenic temperatures, noting that the impact of the Fröhlich interaction on carrier behavior should only increase towards higher temperatures, as LO phonons become thermally available.

We note that the relative coupling strengths for different LO phonon modes are linked to the magnitude of the macroscopic electric field that each mode generates. Furthermore, if the relative contribution of one mode (LO or otherwise) dominates over those of all other phonons, the mode in question may be referred to as a killer mode[22]. This term was introduced for molecular semiconductors, where the phonon mode that contributes most strongly to electron–phonon coupling also drives thermal disorder. However, we do not consider thermal disorder here,

instead defining a killer mode as a mode that dominates carrier–phonon scattering in a certain material. To illustrate the significance of killer mode identification, it is worth noting that, in the context of orthorhombic perovskites like $CsPbBr_3$, the presence of such a mode would lead to great simplifications in the description of carrier–phonon coupling, reducing the number of significant contributions to a single one, from a total 60 normal modes[23]. Note that for an accurate experimental overview of carrier–phonon scattering, all modes must be detectable via the chosen technique. As such, conventional Raman scattering and other related methods with similar selection rules are typically insufficient in perovskites, due to the Raman inactivity of LO phonons in centro-symmetric crystals, which excludes the Fröhlich interaction from the scattering analysis.

Consequently, while carrier–LO phonon coupling via the Fröhlich interaction has been reported in many types of lead halide perovskites[2,3,11,12,24–28], an experimental investigation of the full phonon spectrum has not been carried out thus far. It has been suggested in two separate studies that polaron formation in hybrid perovskites involves mainly low-frequency phonons related to the inorganic sublattice[29,30]. However, the investigated materials had a tetragonal crystal structure, as opposed to the orthorhombic structure of $CsPbBr_3$, such that the relevance of these results in the inorganic species is not immediately apparent. Moreover, both studies employed the technique of femtosecond impulsive stimulated Raman spectroscopy which, once again, cannot directly probe the LO phonons that contribute to the Fröhlich interaction in centro-symmetric materials[31]. The contributions of various low-energy LO phonons to the Fröhlich interaction have been reported in lead halide perovskites[32,33], but no real consensus exists regarding the modes which dominate the interaction. A somewhat definitive answer was reached for $MAPbI_3$, where a single effective LO mode around 15 meV was identified as coupling to electronic motion[34]. Conversely, in $CsPbBr_3$, two lower-energy modes of the lead halide sublattice at 3.7 and 6.3 meV were seen dominating the resonant Raman scattering spectrum at cryogenic temperatures[35]. However, the symmetry of these modes was not well established, and not all of the known normal modes of the perovskite lattice were observed. Moreover, an abnormal decrease of the scattering intensity with increasing temperature introduced additional uncertainty regarding the scattering mechanism responsible for the observed peaks. Further confusion on this topic is generated by the fact that the energies of LO phonons involved in temperature-dependent PL line broadening in $CsPbBr_3$ do not always coincide with the LO phonon energies found in low-temperature PL spectra[36]. Additionally, the former are much larger than the energies of the two modes mentioned above. Consequently, the topic of Fröhlich-active modes in inorganic lead halide perovskites remains open, and requires further analysis via a suitable experimental technique. Note that, under certain conditions, resonant Raman scattering spectroscopy can be used to probe both Raman-active and Raman-inactive modes[37], and is thus suitable for identifying the most significant Fröhlich-active modes in a given material.

We further note that the strength of the Fröhlich interaction in a material is directly linked to the polar nature of its crystal lattice. In a highly polar material the Coulomb field of a carrier (or exciton) couples more easily to the polar vibrations (i.e. LO phonons) of the lattice, resulting in strong Fröhlich coupling. The static and high-frequency values of the dielectric function provide a means of quantifying the polar nature of a material, such that further insight into the Fröhlich interaction can be gained via dielectric characterization.

In this work, we employ a combination of resonant multi-phonon Raman scattering (MPRS) and THz time-domain

spectroscopy (THz-TDS—Supplementary Note 8) to describe the carrier–LO phonon interaction in CsPbBr$_3$ NCs. We record the full phonon spectrum of the material, including both Raman-active and inactive modes. We observe an LO phonon mode that dominates over all carrier–phonon interactions identifying it as a killer mode. Furthermore, the strength of the Fröhlich interaction for the mode in question is quantified and compared to theoretical expectations, given the dielectric response of the material. It becomes clear that the observed low-temperature exciton–phonon interactions cannot simply be due to the polar nature of the lattice. By means of a well-established model, we reveal that these observations can be fully explained by localization which enhances the net charge density of the exciton. Finally, we discuss the unexpected presence of a low-frequency dielectric response of CsPbBr$_3$ at room temperature. We tentatively explain the observation as a universal dielectric relaxation linked to the reorientation of dynamic dipoles which may stem from surface defects. We remark that the Fröhlich interaction in CsPbBr$_3$ nanocrystals (NCs) is representative also for bulk systems and, moreover, that there should be no significant structural differences between the CsPbBr$_3$ lattices at cryogenic and room temperature (Supplementary Note 5). Consequently, our analysis provides a general view of the Fröhlich interaction in CsPbBr$_3$ and related materials.

## Results

CsPbBr$_3$ NCs were synthesized via a hot injection method. Initial characterization was conducted via TEM, as well as PL, and PLE spectroscopy, and the results are shown in Supplementary Fig. 1. The 4 K PL data suggests that the NCs exhibit only weak quantum confinement. We note that, at room temperature and below, the crystal structure of CsPbBr$_3$ is orthorhombic, both in bulk and NC form, whereas all crystal phase transitions occur exclusively above room temperature[2,15,38].

**Multiphonon Raman scattering.** The results of MPRS on CsPbBr$_3$ NCs at 3.8 K are shown in Fig. 1a. The MPRS spectrum shows several intense peaks shifted by near multiples of 155 cm$^{-1}$ (19.2 meV). These represent the multiphonon progression of an LO phonon mode, labeled M$_0$, also observed in previous studies[2,12,39–41]. In the "Discussion" section it will be shown that the observation of mode M$_0$ in this study is strictly linked to the resonant experimental conditions in conjunction with symmetry-breaking factors. As is typical of multiphonon Raman scattering via the Fröhlich interaction[37], the overtone energies were slightly

lower than exact multiples of the phonon energy, each peak exhibiting a red-shift smaller than their respective FWHM. This is a consequence of the forbidden nature of Fröhlich scattering at the center of the Brillouin zone (see "Discussion"), such that the observed peaks are the result of scattering by lower-energy LO phonons of non-zero momentum. The broadening of the overtones with increasing order (Fig. 1a, inset) suggests that the multiphonon spectrum is produced by resonant Raman scattering, as opposed to hot luminescence (Supplementary Note 4). The M$_0$ overtones have an asymmetric lineshape, presenting increased scattering on the lower-energy side. Such asymmetry of Raman scattering peaks in NCs has previously been linked to Fano interference[42], quantum confinement[43], or scattering by lower-energy phonons, related to the NC surface[44,45].

At lower energies in the spectrum, three low-intensity peaks are present, also observed in Raman spectroscopy of single crystal CsPbBr$_3$[15]. The low-energy modes at 28 cm$^{-1}$ (3.5 meV), 53 cm$^{-1}$ (6.6 meV), and 75 cm$^{-1}$ (9.3 meV) are labeled M$_1$, M$_2$, and M$_3$, respectively. For a tentative symmetry assignment of the low-energy modes, we compare the Raman spectrum of CsPbBr$_3$ with the known vibrational spectrum of the orthorhombic perovskite CsPbCl$_3$[46]. When the two materials have the same crystal phase, the phonon modes of CsPbCl$_3$ all have equivalent modes in CsPbBr$_3$. The mode energies and splittings are renormalized according to the differences in ionic mass and octahedral rotation angles, respectively. The renormalized mode energies can be obtained by modeling the phonons as harmonic vibrations of chains of alternating lead and halogen atoms. While the proposed model is quite simple, we find very good agreement between the CsPbCl$_3$ phonon energies adjusted with respect to a change in anion from Cl to Br and the energies of the CsPbBr$_3$ peaks observed in the spectrum of Fig. 1a. We thus tentatively identify M$_1$, M$_2$, and M$_3$ as TO modes for CsPbBr$_3$. Finally, second-order Raman scattering peaks are also observed in the MPRS spectrum. The second-order peaks involve combinations of mode M$_0$ and each of the three other modes (correspondingly marked M$_0$ + M$_1$, M$_0$ + M$_2$, and M$_0$ + M$_3$ in Fig. 1a).

Before proceeding we must note that the CsPbX$_3$ lattice also supports LO modes at energies below M$_0$[32,46,47]. These additional modes should be energetically close to the non-polar TO modes M$_1$, M$_2$, and M$_3$, and may therefore be hidden in the measured spectrum. However, strong Fröhlich scattering comparable to that involving mode M$_0$ should still produce visible overtones of these hidden modes. In the MPRS spectrum, no multiphonon

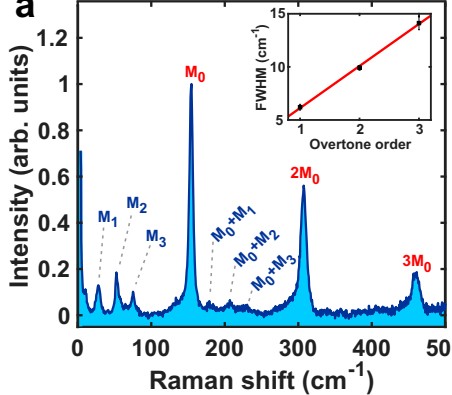

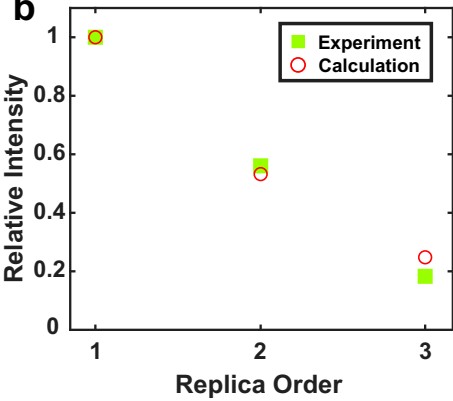

**Fig. 1 MPRS results for CsPbBr$_3$ nanocrstals (NCs). a** MPRS spectrum of CsPbBr$_3$ NCs at 3.8 K. The labels represent the different phonon modes observed and their combinations. The inset shows an increase of the FWHM for consecutive overtones. The FWHM values and their respective errorbars were obtained via Gaussian fitting of each peak. The red line is a guide to the eye. **b** Experimental and calculated intensities of the M$_0$ replicas in **a**. Calculations assume $S = 0.39$, $\sigma = 32.5$ meV, $\Gamma_0 = 1$ meV, and $E_g^0 = 2.375$ eV. The maximum relative measurement error corresponds to the third-order replica, and is around 10%, due to the subtraction of the PL background. The sum of squared residuals for the fit is 0.005.

progressions are observed in the low-energy range, meaning that $M_0$ must be the dominant contributor to the Fröhlich interaction in CsPbBr$_3$. Furthermore, since scattering by $M_0$ clearly dominates the MPRS spectrum, which contains contributions from both Raman-allowed and forbidden modes, $M_0$ in fact represents the strongest overall contributor to carrier–phonon scattering (Fröhlich or otherwise) in CsPbBr$_3$, and thus behaves as a killer phonon mode in the material.

**Terahertz time-domain spectroscopy.** In order to quantify the polar nature of CsPbBr$_3$, THz-TDS analysis was used to determine the frequency-dependent real and imaginary dielectric function of CsPbBr$_3$ NCs at room temperature. The results are shown in Fig. 2a. A trusted frequency range, with a strong signal-to-noise ratio, is defined between 0.2 and 1.7 THz (Supplementary Fig. 7). The experimental data are modeled with a modified Debye–Lorentz dependence, where the first term corresponds to a low-frequency dipolar relaxation[48,49] and the second term represents a lattice resonance:

$$\varepsilon(\omega) = \frac{\varepsilon_S - \varepsilon_\infty^{CD}}{(1 - i\omega\tau)^\beta} + \frac{\varepsilon_\infty^{CD} - \varepsilon_\infty}{\omega_0^2 - \omega^2 - i\gamma\omega} + \varepsilon_\infty. \tag{1}$$

Here $\varepsilon_S$ and $\varepsilon_\infty$ are the static and high-frequency dielectric constants, while $\varepsilon_\infty^{CD}$ is an intermediate value of the dielectric function, which would constitute the high-frequency value in the absence of the Lorentz term. The Debye-like term is known as a Cole–Davidson dependence, and contains a skewing factor $\beta$ which accounts for a distribution of relaxation times $\tau$. Finally, $\omega_0$ and $\gamma$ represent the Lorentzian resonance frequency and damping factor, respectively. As shown in Fig. 2b, the standard Debye model fails to accurately reproduce the low-frequency dielectric function. The Cole–Davidson skewing factor $0 < \beta < 1$ stretches out the relaxation towards higher frequencies, producing a better fit of the experimental data than the Debye model.

Note that the measurement data do not cover the entire frequency range of the dielectric relaxation, such that the time constant $\tau$ is not well-defined. To more accurately determine the relaxation time, data points corresponding to lower frequencies must be added. To this end, the experimentally obtained dielectric function of CsPbBr$_3$ in the MHz range is used here. The value of the real dielectric function at 1.08 MHz, of $\varepsilon'_{1.08\text{MHz}} = 20.5$ (ref. [50]), is added to the measurement data, thereby obtaining $\tau = 32 \pm 1$ ps, and a skewing factor $\beta = 0.61 \pm 0.01$ (Fig. 2a).

The low-frequency relaxation is not observed in the reference measurement of the solvent (Supplementary Fig. 9) and is thus definitively attributed to the presence of CsPbBr$_3$ NCs in the measured sample. The Lorentz resonance is situated at 2.02 ± 0.01 THz. While this lies outside the trusted frequency range, the feature is also convincingly attributed to the presence of CsPbBr$_3$ NCs in the sample solution (Supplementary Fig. 9). Moreover, the corresponding energy is close to an optical phonon energy of CsPbBr$_3$, at 70–80 cm$^{-1}$ (2.1–2.4 THz)[15].

From our model we obtain the static and high-frequency dielectric constants $\varepsilon_S = 20.5 \pm 0.2$ and $\varepsilon_\infty = 4.8 \pm 0.1$. The value of $\varepsilon_\infty$ is consistent with previous calculations[3,8,51] and experimental results[52], while $\varepsilon_S$ is fixed by the experimental value at 1.08 MHz[50].

## Discussion

In interpreting the MPRS results, we note once again that non-resonant Raman scattering involving Fröhlich-active (i.e. polar) LO modes (like $M_0$) is symmetry-forbidden in crystals with a center of inversion, such as orthorhombic lead halide perovskites. Specifically, while modes $M_1$, $M_2$, and $M_3$ are non-polar TO phonons which are Raman-active, $M_0$ scattering remains negligible under non-resonant conditions[15]. However, under resonant conditions, the energy of incoming photons corresponds to that of excited electronic states, which couple to LO phonons through the Fröhlich interaction. Strong forbidden multiphonon scattering involving LO phonons thus becomes possible in resonant Raman experiments, thereby explaining the high scattering intensity of the peaks involving mode $M_0$ in our MPRS spectrum[37]. Note that, even at resonance, Raman scattering by a single LO phonon must obey the condition $\mathbf{k} \simeq 0$, and remains nominally forbidden. However, this restriction is lifted by symmetry-breaking factors[53]. One commonly considered factor are defects, which relax crystal momentum conservation, allowing scattering to occur away from the center of the Brillouin zone. As shallow trapping of carriers has indeed been reported in CsPbBr$_3$ NCs[2,54,55], defects may well be the reason for the occurrence of first-order forbidden scattering by $M_0$ phonons in our experiment.

A quantitative analysis of Fröhlich scattering in CsPbBr$_3$ can be obtained by considering the relative cross-sections of the $M_0$ overtones. The relative cross-sections are estimated through a configuration-coordinate model[44,53,56] (Supplementary Note 6). The $n$-phonon Raman cross-section at low temperature is then[44]:

$$\sigma^{(n)} = \mu^4 \left| \sum_{m=0}^{\infty} \frac{\langle g, n | e, m \rangle \langle e, m | g, 0 \rangle}{E_g + m\hbar\omega_0 - \hbar\omega_L + i\Gamma_0} \right|^2, \tag{2}$$

where $\mu$ is the dipole moment of the electronic transition, $E_g$ is the excitonic transition energy, $\hbar\omega_L$ is the energy of incident photons, $\hbar\omega_0$ is the LO phonon energy, and $\Gamma_0$ is the single-NC PL linewidth. The brackets $\langle e, u | g, v \rangle$ are overlap integrals between

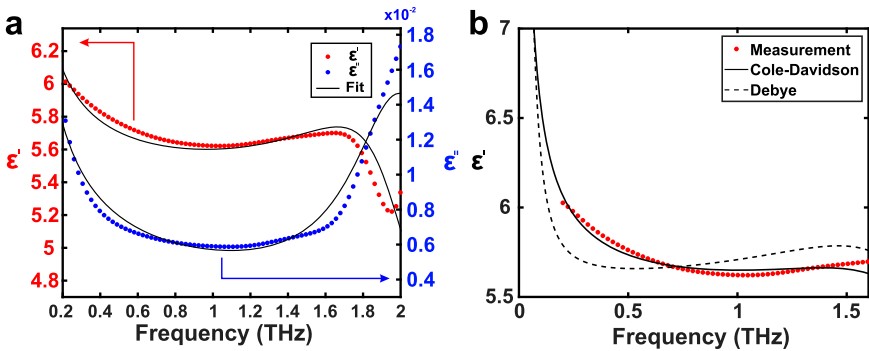

**Fig. 2 THz-TDS results for CsPbBr$_3$ NCs dispersed in hexane. a** Measured real (red) and imaginary (blue) dielectric function of CsPbBr$_3$ NCs from THz-TDS. The data are fitted with Eq. (1). **b** Comparison of the Debye and Cole Davidson model for the real dielectric function. In this subfigure, both models are restricted to the low-frequency region, thereby excluding the Lorentz resonance. The Cole–Davidson fit corresponds to $\tau = 21$ ps and $\beta = 0.71$, giving a sum of squared residuals of 11.5, compared to 77.8 for the Debye model. Errors in the THz-TDS analysis are discussed in Supplementary Note 7.

vibrational levels of the ground and excited electronic states, and are defined for $u \geq v$ as

$$\langle e, u | g, v \rangle = e^{-S/2} \sqrt{u! v!} S^{\frac{u-v}{2}}$$
$$\sum_{j=0}^{v} \frac{(-S)^j}{(v-j)!(u-v+j)!j!}, \qquad (3)$$

while for $u < v$, the identity $\langle e, v | g, u \rangle = (-1)^{u+v} \langle e, u | g, v \rangle$ holds. In the above equation, $S$ is the so-called Huang–Rhys factor, which is used to quantify the strength of carrier–phonon coupling for a given mode (Supplementary Information, Section 5).

The relative intensities of the $M_0$, $2M_0$, and $3M_0$ lines are estimated using Eq. (2), which is integrated over a Gaussian distribution of $E_g$ (Supplementary Note 6), with a variance $\sigma$ comparable to that of the 4 K PL lineshape. The parameters used in the calculation are given in Table 1.

The low-temperature PL linewidth of single CsPbBr$_3$ NCs is typically $\Gamma_0 < 2$ meV[57,58], and thus the assumption of $\Gamma_0 \simeq 1$ meV is reasonable. The Huang–Rhys factor, $S = 0.39$, is obtained via nonlinear least-squares regression for the relative intensities of the $M_0$, $2M_0$, and $3M_0$ peaks, and excellently reproduces the experimental progression, as shown in Fig. 1b. This value of $S$ is in good agreement with previous reports[2,10]. Note that any choice of the estimated parameters within the experimental uncertainty produces the same order of magnitude of $S$ (Supplementary Fig. 4). Finally, we note that one previous study has shown that phonon replicas may appear in the low-temperature PL spectrum of CsPbBr$_3$ NCs with a similar Huang–Rhys factor[2]. We remark however that the linewidth of our PL emission (Supplementary Fig. 1) is twice as large as that in the respective reference. Since phonon replicas are only visible if the linewidth is sufficiently low in comparison to the phonon energy, a doubling of the linewidth can thus completely erase any trace of multiple PL bands, even for much higher $S$ values than the one found here[59].

By definition, the Fröhlich interaction couples the Coulomb field of a carrier with the macroscopic electric field of LO phonons, forming a lower-energy, heavier quasiparticle called a polaron. The intrinsic carrier–LO phonon coupling strength is given by the Fröhlich constant[60]:

$$\alpha = \frac{e^2}{\hbar \omega_0} \left( \frac{1}{\varepsilon_\infty} - \frac{1}{\varepsilon_S} \right) \sqrt{\frac{m \omega_0}{2\hbar}}, \qquad (4)$$

which determines the self-trapping energy and enhanced mass of a carrier of effective mass $m$ interacting with the lattice[61,62]. The quantity $\frac{1}{\bar{\varepsilon}} = \left( \frac{1}{\varepsilon_\infty} - \frac{1}{\varepsilon_S} \right)$ is known as the dielectric contrast, and effectively quantifies the ionic nature of a material, thereby determining the strength of the carrier–lattice interaction. Using experimental values for $\varepsilon_S$, $\varepsilon_\infty$, and $\hbar\omega_0$, along with the carrier effective masses given by Yettapu et al.[51], of $m_e = 0.22$ and $m_h = 0.24$, the Fröhlich constant in CsPbBr$_3$ is $\alpha \simeq 2$ for both electrons and holes. While in the case of excitons the reduced

mass is sometimes used to calculate the Fröhlich constant, this approach does not provide an accurate picture of the behavior and physics of excitonic polarons. Still, any argument regarding the strength of electron–phonon coupling remains qualitatively valid for exciton–phonon coupling as well, keeping in mind that the latter is likely weaker than the former, due to the compensation of the electron and hole charges upon exciton formation, which diminishes their respective phonon clouds. We further note that the Huang–Rhys factor is directly proportional to the Fröhlich constant (Supplementary Note 6). Additionally, the magnitude of the Huang–Rhys factor is determined by a summation over all available phonon wave vectors[63]. In the case of free electron or hole polarons, the phonon occupation is determined by the thermal energy and polaron mass (Supplementary Eqs. (4) and (5)). At the 3.8 K corresponding to our MPRS experiment, to reach the value $S = 0.39$, the low thermal energy must be compensated by an absurdly large polaron mass, more than 20 times the bare electron mass. This would require $\alpha \gg 1$[60,62,64]. For an excitonic polaron, an even greater mass enhancement is likely necessary, as suggested above. For CsPbBr$_3$, the value of $\alpha \simeq 2$ corresponds to an intermediate carrier–lattice coupling regime, where the effective mass correction remains relatively small. Cyclotron resonance measurements and estimations of the bare carrier mass via density functional theory suggest that the polaron is only ~50% heavier[21,51,65,66], which is insufficient to account for the measured Huang–Rhys factor, instead resulting in $S \simeq 0.07$.

Note that the requirement of an unrealistic carrier mass is unlikely to result from a faulty estimation of $\varepsilon_S$ and $\varepsilon_\infty$, since the values used here are reasonably close to previous reports. Furthermore, the polaron mass enhancement is linearly dependent on $\alpha \propto \bar{\varepsilon}^{-1}$, which is a relatively slowly-varying function of the two dielectric constants. Finally, we recall from the introduction that anharmonicity effects may influence carrier–phonon scattering in lead halide perovskites[15–18]. In fact, the anharmonic nature of the CsPbBr$_3$ lattice at room temperature may even be linked to strong self-trapping of excitonic polarons[67]. However, strong anharmonic contributions at room temperature are known to give rise to a strong central peak in the Raman spectrum of a material—an effect that has not been observed at cryogenic temperatures in lead halide perovskites, neither in previous experiments below 77 K[15,68,69], nor in our own experiment, presented here. We can therefore surmise that the lattice anharmonicity is significantly reduced at the low temperature of our experiment (3.8 K), and thus the $S$ factor cannot be intrinsically accounted for.

We therefore look for extrinsic factors that may enhance the Fröhlich interaction. The occurrence of first-order forbidden LO phonon scattering in our MPRS experiment is thus very telling, as it suggests that the exciton–phonon interaction involves a breaking of the lattice symmetry, likely in the presence of defects. Models involving defect-localized charges have been used extensively to explain strong Fröhlich interactions in nanostructures[44,56,70–74], as well as in the bulk[53,63]. Localization is also known to enhance Fröhlich coupling in bulk CsPbBr$_3$[75], thus supporting the implementation of such a model in describing our observations. The broken lattice periodicity at a defect enables the relaxation of wave vector conservation rules, thereby enabling carrier–phonon coupling with $\mathbf{k} > 0$[63]. In this case, the Huang–Rhys factor is determined by the extent of the exciton charge distribution in Fourier space, which is enhanced by the trapping potential (Supplementary Note 6). To estimate the Huang–Rhys factor in the case of a defect-bound exciton, a model charge distribution may be used, where one carrier is pinned, while its counterpart is delocalized over the NC[44,56,71,72]. The accessible volume in Fourier space is then defined by the average NC size (~11 ± 3 nm), and the Huang–Rhys factor is

**Table 1 Parameters used in the calculation of the relative scattering cross-sections shown in Fig. 1b.**

| | |
|---|---|
| $E_g$ (eV) | 2.375* |
| $\sigma$ (eV) | 0.0325* |
| $\hbar\omega_L$ (eV) | 2.54 |
| $\hbar\omega_0$ (eV) | 0.0192 |
| $\Gamma_0$ (eV) | 0.001* |
| $S$ | 0.39 |
| RSS | 0.005 |

Values marked with an asterisk represent estimates based on experimental evidence from this work ($E_g$, $\sigma$), or previous reports ($\Gamma_0$). RSS is the residual sum of squares obtained with the given parameters.

$S = 0.42 \pm 0.11$, in excellent agreement with our experimental result. We therefore argue that exciton localization can fully explain the strong low-temperature Fröhlich interaction in $CsPbBr_3$.

It must however be emphasized that $M_0$ is in fact a normal mode of the orthorhombic perovskite lattice, meaning it does not arise due to any distortion generated by defects, but occurs naturally in the pristine material as well. Furthermore, exciton localization modifies only the carrier contribution to the Fröhlich carrier–lattice interaction (Supplementary Note 6), meaning that the macroscopic electric field of each LO phonon remains unaffected. That means that a free carrier (or, indeed, any other type of carrier) will also interact most strongly with mode $M_0$, out of all available LO modes. The dominance of $M_0$ over the Fröhlich interaction is thus not exclusively defect-related instead reflecting the intrinsic properties of the $CsPbBr_3$ system.

Finally, the low-frequency dielectric response of $CsPbBr_3$ has significant influence on carrier transport, even beyond the Fröhlich interaction. This is because dielectric relaxations below the lattice resonance frequency are attributed to dipolar reorientation and ionic motion, leading to carrier screening, which affects mobility.[6,7,76] In Fig. 2a, we observed a relaxation in the sub-THz range, which results in a doubling of the dielectric loss over less than one decade of frequency. A similar relaxation has been measured in the hybrid perovskite $MAPbBr_3$, and it has been attributed to the rotation of $MA^+$ cations, which possess an intrinsic dipole moment[6,7,76]. It has been suggested that a similar response should be observable in the fully inorganic $CsPbBr_3$, even in the absence of rotating dipoles, the response arising instead from the rattling motion of the $Cs^+$ cations in their atomic lattice positions[6]. Indeed, collective displacements of lattice ions in many ionic materials lead to so-called configurational tunneling, which produces what has been referred to as a universal dielectric response below the lattice frequency[48,49]. However, recent results on the dielectric response of single crystals[77] suggest that such a relaxation is completely absent in fully inorganic halide perovskites, while our measurements clearly show that a low-frequency response is in fact present in $CsPbBr_3$ NCs. The absence of the relaxation in bulk single crystals indicates that this response cannot be related to the $Cs^+$ ion motion. We therefore propose that the low-frequency relaxation we observed is produced by configurational tunneling involving a different entity. The response may, for example, be associated with defects, related to the NC surface. Note that most defects in the perovskite lattice have a favorable energetic position close to the edges of the valence and conduction bands[78], and do not increase the rate of nonradiative recombination. Therefore the proposition that defects may contribute to the observed dielectric response does not contradict the excellent PL efficiency of $CsPbBr_3$ NCs. The link between the dipolar relaxation and lattice defects in $CsPbBr_3$ is supported by molecular dynamics simulations, which have shown that the positions of lattice ions at halide vacancies fluctuate on a similar time scale (5 ps)[79] as our measured relaxation ($\tau = 32$ ps) (by comparison, relaxation due to $Cs^+$ motion occurs on a time scale of around 100 fs[15]). The charge density around a defect depends strongly on the positions of neighboring ions, such that these fluctuations may induce a dipole which can respond to an electromagnetic field, as required for configurational tunneling. We further note that in the current scenario, the analysis was conducted in such a way as to extract only the response of the perovskite lattice to the applied field (Supplementary Note 8). That is, the organic ligands were excluded from the volume fraction of the NC inclusions. Consequently, the dipolar relaxation is not attributable to the ligands. Finally, we note that the possible involvement of defects in the enhancement of the Fröhlich interaction as well as the generation of a dielectric response does not imply that the two effects are inter-related.

This is immediately apparent when considering that the first mechanism describes the effects of localization on the carrier–phonon interaction, while the defect itself may be substituted by other symmetry-breaking factors. Conversely, the second mechanism is strictly related to the dynamics of defect-neighboring ions, and not the carriers themselves.

In conclusion, we have shown that the LO phonon mode around 20 meV acts as a killer mode, being chiefly responsible for carrier–phonon scattering in $CsPbBr_3$. The low-temperature Huang–Rhys factor $S \simeq 0.39$ was determined from the relative resonant Raman cross-section of consecutive overtones. The dielectric response of $CsPbBr_3$, investigated by means of THz-TDS, indicates that the intrinsic strength of the Fröhlich interaction is incompatible with the low-temperature Huang–Rhys factor. We have shown that localization may enhance carrier–phonon coupling sufficiently to explain this experimental observation. As the Huang–Rhys factor obtained here falls within the typical range of reported values for lead halide perovskites, it thus becomes necessary to consider that such enhancements may in fact play a role in many reported cases of carrier–LO phonon coupling in this class of materials.

Concerning the significance of mode $M_0$ in the context of polaron formation, we recall the recent report that the self-trapping of excitons in $CsPbBr_3$ NCs involves two lower-energy phonon modes[35], which seems to contradict our results. We note however that the spectrum recorded in the reference was likely incomplete, due to the absence of the resonance with mode $M_0$. We hereby underscored the importance of recording the full phonon spectrum in order to properly describe carrier–phonon interactions in any material, as an incomplete analysis can produce misleading conclusions. In the current study we managed to alleviate the existing confusion in the field by recording the full phonon spectrum of $CsPbBr_3$, including both Raman-allowed and Raman-forbidden modes over a broad energy range. While scattering via $M_0$ has also been observed, to a certain degree, in other experiments[15,40], the forbidden nature of LO phonon scattering resulted in a weak Raman signal and could not provide any indication of the strength of the Fröhlich interaction. By contrast, our experiment was done under resonant conditions, and the symmetry-related selection rules were broken. This fully enabled the observation of forbidden Fröhlich scattering, even in the first order, thereby allowing the quantification of the strength of carrier–lattice interaction. As such, we were able to unambiguously identify $M_0$ as the main contributor to the Fröhlich interaction, and thereby polaron formation and scattering in $CsPbBr_3$. This fact should henceforth be considered in the interpretation of scattering-related experiments and polaronic phenomena in this material and likely other perovskite species with an orthorhombic structure.

Additionally, the low-frequency dielectric response of $CsPbBr_3$ merits consideration. Here we tentatively linked the observed Debye-like dielectric relaxation to surface defects, due to the low probability of finding internal imperfections under a NC geometry. Presumably, a similar response can arise from a wide range of defects occurring in the $CsPbBr_3$ lattice. Consequently, defect-prone materials such as polycrystalline thin films may present a similar dielectric relaxation, which would have an impact on carrier transport, by modifying the amount of carrier screening from defects. A more focused analysis should be conducted to determine whether such a relaxation does in fact occur in other systems, whether its effect on carrier transport is considerable, and whether this effect is positive or negative.

Ultimately, different applications require different material characteristics, and improving the performance of any device is fully reliant on a good understanding of the basic properties of the materials used. For the intended optoelectronic applications of $CsPbBr_3$, this involves knowledge of carrier relaxation, recombination, and transport, all of which are inextricably linked to the Fröhlich interaction. As such, the defining features of this

interaction identified here constitute a step towards the full understanding of CsPbBr$_3$, and possibly other perovskite species, in the interest of developing novel optoelectronic devices.

## Methods

**Materials used for CsPbBr$_3$ NC synthesis**. Cesium carbonate (Cs$_2$CO$_3$, Sigma-Aldrich, 99.9%), 1-octadecene (ODE, Sigma-Aldrich, 90%), oleic acid (OA, Sigma-Aldrich, 90%), lead bromide (PbBr$_2$, Sigma-Aldrich, 99.999%), oleylamine (OLAM, Sigma-Aldrich, 70%), anhydrous toluene (Sigma-Aldrich, 99.8%), and anhydrous hexane (Sigma-Aldrich, 95%) were used. OLAM and OA were degassed under reduced pressure at 120 °C for 1 h prior to use. All other chemicals were used as received.

**Sample fabrication and measurements**. NCs were synthesized via the reaction of Cs-oleate with PbBr$_2$ in octadecene at 140–200 °C. The NCs were stabilized in suspension by adding a 1:1 mixture of oleylamine and oleic acid. The obtained solution was centrifuged for 3 min at 10,000 r.p.m. and the colored supernatant was discarded. Then, 300 μL of hexane was added and the NCs were dispersed using a vortex mixer. Subsequently, the suspension was again centrifuged for 3 min at 10,000 r.p.m., after which the precipitate, containing larger NCs and agglomerates, was discarded. Another 300 μL of hexane was added to the supernatant, resulting in a colloidal dispersion of CsPbBr$_3$ NCs. Raman scattering measurements were performed at the High Field Magnet Laboratory (Radboud University). To achieve low temperatures, samples were placed in a liquid $^4$He optical cryostat (Microstat Oxford). Samples were probed in backscattering geometry with an incident laser line at 2.54 eV (488 nm) from a solid-state laser, well above the low-temperature NC emission energy at 2.38 eV (521 nm). The scattered light was filtered by a RazorEdge ultra steep long-pass edge filter and analyzed by a 1 m long single grating (1200 gr/mm) FHR-1000 Horiba spectrometer equipped with a nitrogen-cooled PyLoN CCD camera (Princeton Instruments). Dielectric spectroscopy was carried out via THz-TDS at room temperature, on a colloidal CsPbBr$_3$ NC sample dispersed in hexane, with an average NC size of 9.5 ± 1.3 nm. A focused THz wave packet was used to probe a high-density colloidal suspension of CsPbBr$_3$ NCs, and the frequency-dependent transfer function of the sample was analyzed in order to extract the complex refractive index $\tilde{n} = n + i\kappa$, where $n$ is the real refractive index and $\kappa$ is the extinction coefficient. The technique is described in detail in Supplementary Note 8.

## Data availability

The data that support the findings of this study are available from the corresponding author upon reasonable request. Source data are provided with this paper.

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

## Acknowledgements

This work is part of the research program entitled "Designing Dirac Carriers in Semiconductor Superlattices" (DDC13), which is supported by the Foundation for Fundamental Research on Matter (FOM) which is part of the Netherlands Organization for Scientific Research (NWO).

## Author contributions

The idea for the manuscript is the result of continuous interaction between C.M.I. and A.Y.S.; NC samples were fabricated and characterized by F.M. and S.B., while A.B. and J.B. conducted the MPRS experiment; C.M.I. conducted the THz-TDS measurements with the help and supervision of N.J.J.v.H. and S.E.T.t.H.; C.M.I. analyzed the experimental data and wrote the manuscript with input from A.B., N.J.J.v.H, S.E.T.t.H., J.B., F.M., S.B., P.C.M.C., D.V., C.d.M.D, J.G.R., P.M.K., and A.Y.S.; A.Y.S and P.M.K. directly supervised the project.

## Competing interests

The authors declare no competing interests.
