## [Peer Review File · Nature Communications]

REVIEWER COMMENTS

Reviewer #1 (Remarks to the Author):

The authors performed Raman spectroscopy on nanocrystals of CsPbBr₃ perovskite and identified a resonant mode at around 20 meV (along with their replicas) to be dominant phonon mode coupled to electronic excitations. They performed a rigorous analysis of the Raman intensities of the observed Raman lines to estimate the Huang-Rhys factor which represent the degree of displacement of the electronic PES along the phonon co-ordinates in the presence of electronic excitations. The authors argue that the HR factor of 0.4 is too high to be expected for a large polaron with a polaron coupling coefficient of 2. The authors suggest that this mode is an outcome of carrier localization at a defect – a self-trapped exciton and not free large polaron. Independently, the authors also perform THz-TDS spectroscopy and estimate the dielectric relaxation rates, which again is speculated to be a consequence of defects. The observed LO phonon energy is well known in the literature, while the Huang-Rhys factor is a much-needed quantitative addition. The discussion on its origin from defects is however not well developed and needs major revision to be considered for publication in Nature Communications.

My specific concerns are as follows:

1. The authors should define better the term “killer” mode which is not a commonly used terminology. The provided reference is in the context of molecular semiconductors where the discussion is about modes that drive thermal disorder. Do the authors imply that the mode identified by them is the driver of disorder in the perovskite lattice? What is the evidence of the same?
2. The authors should elucidate better why $S \sim 0.4$ is not consistent with large polaron model in the main manuscript. This is the crucial component of the manuscript and the discussion is mostly relegated to the SI, with most of the discussion there on self-trapped excitons. I understand that the large Huang-Rhys factor does indicate strong coupling to phonons, but why should it lead to the seven-fold enhancement in the effective mass? This is not clear from any of the equations presented in Section S5 of the SI. An important reference in this context is the work from Mathies and co-workers (Nature Communications, 9, 2525(2018)), where they theoretically estimated the displacement of PES and experimentally measured it. They suggest that this is intrinsic characteristic of metal halide perovskites mainly dictated by the inorganic lattice component. Similar observations were made by Batignani et al in Nature Communications, 9, 1971 (2018).
3. Let us consider that the mode M0 is indicative of carrier localization at defects, meaning it is the mode along which the lattice re-organizes in the presence of trapped charges in the lattice. Why

should it have any relevance in free carrier transport or hot carrier thermalization. In the case of transport, does M_0 mode play a role in de-trapping mechanism? There is however no evidence of any such de-trapping processes or a role of phonons, at least to my knowledge.

4. With such large S , one should be able to observe vibronic replicas in the linear absorption and PL spectra, is it the case? Of course, if this is related to self-trapped exciton, then one might expect to see the replicas exclusively in the emission spectra (at low temperatures).

5. The authors also claim that the THz response, which is indicative of dielectric relaxation, is also defect mediated. Firstly, it has been suggested and partly proven that CsPbBr₃ NCs are almost defect free, as indicated by their high PL quantum yields. Even in the presence of defects, which may be of the order of $10^{14-15} \text{ cm}^{-3}$, which is one in a hundred thousand unitcells, in this case less than 0.1 defects per nanocrystal. Why should such small density of defects dictate the dielectric relaxation? I agree that the slow picosecond dielectric relaxation timescales are striking, but I am not convinced that this is a result of defects. Is there a role played by the organic ligands that cap the nanocrystal?

Reviewer #2 (Remarks to the Author):

The manuscript by Iaru et al. addresses the Fröhlich mechanism in halide perovskite (HaP) CsPbBr₃ by means of multiphonon Raman scattering and THz time-domain spectroscopy. An LO phonon mode at 155 cm^{-1} , the M_0 mode, is reported to dominate Fröhlich interaction in CsPbBr₃ nanocrystals (NCs). From the relative intensities of the M_0 mode and its overtones, a Huang-Rhys factor, $S=0.39$, could be determined. The authors contrast this value to their calculations of the intrinsic value of the Huang-Rhys factor and suggest that the measured one is therefore not intrinsic. Charge localization due to defects has then been suggested to explain the experimentally measured Huang-Rhys factor.

The reviewer thinks that experimental work and analysis appear to be carefully done, with the presented data supporting the importance of the M_0 phonon mode (cf. Fig. 2a) and the emergence of a small Huang-Rhys factor in CsPbBr₃ NCs (cf. Fig. 2b). As such, the work is interesting and solid. However, the reviewer has a number of fundamental concerns regarding the interpretation and hypothesis put forward on the basis of the experimental findings, as follows:

1. Failing of the Fröhlich model: the authors entirely neglect the possibility that the Fröhlich model is failing here. To see why this is relevant, consider that the Fröhlich model describes the interaction of one electron with one LO phonon. Thus, the model is based on the harmonic approximation which is not valid for HaPs at the experimental conditions of this work (see, e.g., Yaffe et al. PRL 118, 136001

(2017), Gold-Parker et al. PNAS 115, 11905–11910 (2018), Gehrman and Egger Nature Communications 10, 3141 (2019), Lahnsteiner and Bokdam arXiv:2101.06099 [cond-mat.mtrl-sci] (2021)). If the assumption is invalid, why should the model work? Isn't it at all likely the mode is incorrect and HR still intrinsic? This should be discussed.

2. Interpretation as defect localization: for easy reading the value for the Huang-Rhys factor obtained from the Fröhlich value should be given in the text, providing whether the Fröhlich model, using the parameters provided in the manuscript, underestimates or overestimates the measured Huang-Rhys factor. Also, it will be interesting to compare the obtained HR value to the literature reporting S for defect transitions via multiphonon recombination within the Fröhlich model, e.g., by Kirchartz et al., J. Phys. Chem. Lett. 9, 939–946 (2018), for the closely related hybrid HaPs. There, using similar physical models S is MUCH larger ($S \sim 10$). How can this be reconciled with the notion and interpretations presented herein?

3. Discussion regarding dielectric drag: the discussion with regard to dielectric drag etc. is shallow and not insightful. It is entirely unclear to the reviewer what “dielectric drag” even means, are the authors talking about polarons? What has ever been predicted or explained or tested experimentally that is suggesting dielectric drag even exists? Or that it is a meaningful scientific theory (allowing for deductions that bear substance and are testable by experiment). The reviewer is aware of the partially obscure literature in the context of HaPs and “dielectric drag”, but finds this notion entirely confusing, especially when discussed side-by-side with the well-established polaron concept. The reviewer suggests removing or at least mellowing this discussion.

4. Discussion about polaron mobility: the present article is alluding quite heavily to polaron mobility. This is a stretch, since the authors do not present any data or analysis that suggests this has even been studied here. Of course it is fine to mention it but perhaps not in such a prominent manner.

Reviewer #3 (Remarks to the Author):

This work by Iaru et al. examines phonon modes in CsPbBr₃ nanocrystals, finding that a single phonon mode exhibits by far the strongest activity through the Fröhlich interaction. This strength of this interaction is found by numerical analysis of experimental data, and a relaxation in the dielectric constant is explained in terms of dielectric drag.

This paper represents a convincing argument in favour of its title, that the Frohlich interaction is dominated by a single phonon mode in the material studied. However, it currently has deficiencies in its clarity, narrative strength and significance for the field.

The Results section starts by introducing the acronym 'NCs' for nanocrystals, without defining it. In itself this is of course only a minor stylistic error, but it underlines the fact that no reference is made to the study being about nanocrystals in the introduction. Lead halide perovskites have indeed achieved remarkable success in recent years in many applications, but their standout achievements have been in the bulk form in thin-film solar cells. Nanocrystal behaviour is not always the same as bulk behaviour (see e.g ten Brinck et al. *ACS Energy Lett.* 2019, 4, 2739–2747, Akkerman et al *Nat. Mater* 2018, 17, 394-405) though of course there will be considerable overlap. The introduction and later part of the paper should therefore be modified to clarify that the study was performed on nanocrystals, and to justify whether or not the results can be applied to and compared to bulk CsPbBr₃.

On a related note, Raman measurements on single crystals are used to justify the attribution of certain Raman peaks to TO phonons on line 66, but the Yaffe et al. paper cited does not give much attention to phonons except in first-principles DFT calculations in the supplementary material. This argument thus needs either more clarification, or strengthening from other sources.

Other points at which additional clarity (or stylistic improvement) is needed in the results section include:

-line 47: at which low temperatures do CsPbBr₃ NCs have an orthorhombic crystal structure? In general, the temperatures at which the measurements were taken need to be made clearer, and the term 'low temperature' better defined at least once in the paper.

-line 49: what are the respective bandgap values?

-Figure 1d: what temperature does this crystal lattice represent? How was it generated, or is it merely an illustration of a generic orthorhombic perovskite lattice? This subfigure does not seem to add to the narrative.

-Line 54: What is the uncertainty in the 19.2 meV shift? Was it not possible to observe further overtones - a straight line fit to only three data points is not very convincing.

-line 55: A reference to Ramade et al would be a good addition: *Appl. Phys. Lett.* 112, 072104 (2018);

-Figure 2b: the legend should read 'Calculation' or 'Theory' for the red data, and the 'relative cross-sections' legend in the bottom left should be eliminated as it duplicates the data points in the figure itself

-Line 78: what is the nature of the 'carriers' being considered? Do they mean 'charge-carriers'? In which case, are they excitons or free-carriers?

On page 7, the usage of an additional dielectric function data point from a separate study needs additional clarification. Was this value obtained by similar methods? If not, it should be justified why it is appropriate to mesh it with the data in this paper since no one method measures the 'true' value of the dielectric function, and there is often large variation depending on the technique used and form of the material. Additionally, the value of 20.5 in reference 32 seems to have been measured at a different temperature from the measurements in this paper (although as mentioned above, the temperature at which measurements were taken in this paper is conspicuously opaque) - whether this temperature difference matters needs to be discussed. Finally, the fact that the ϵ_S value obtained from the fit is the same as that inputted from reference 32 needs discussion: it is dubious to note that ' ϵ_S is the same as the experimental value at 108 MHz' as if to strengthen the credibility of the value obtained from the fit, when that very experimental value was inputted into the fit in the first place!

Secondly, the narrative of the paper is rather lacking. The results themselves constitute good data, from which valid conclusions are drawn. However, the results section is presented as a list of the experiments done, without motivation. For example, on line 51: why are further investigations carried out using these techniques in particular? Figure 1 in general does not add much to the narrative and could be left in the SI without loss to the paper, other than that there would only be one figure left. In the discussion, there should at least be an introductory sentence justifying the upcoming analysis and why it follows from the results, though the discussion itself is convincing.

This brings me to my final contention, namely the significance of the paper to the field. Within the study of CsPbBr₃ nanocrystals, it is not entirely clear from the discussion how much of their findings are novel, as opposed to just corroborating previous findings. Perhaps this is just a consequence of the aforementioned weak narrative, however - although their numerical values of S and M_0 seem to be backed by previous studies, their interpretation of their significance appears novel. Nonetheless, this aspect of the paper needs to be strengthened. Within the field of lead halide perovskites as a whole however, the significance is less clear still. While the examples given in the conclusion of the relevance of LO phonons to the functionality of devices are true, it is less apparent how their findings about the dominant phonon mode in CsPbBr₃ nanocrystals at low temperature are relevant to the behaviour of devices at room temperature. It needs to be clarified how precisely the fact that M_0 'effectively controls carrier dynamics in CsPbBr₃' could be of practical use: just because phonons are involved in a process, it does not immediately follow why particular knowledge of them is useful.

Reviewer 1

Comment 1. *The authors should define better the term “killer” mode which is not a commonly used terminology. The provided reference is in the context of molecular semiconductors where the discussion is about modes that drive thermal disorder. Do the authors imply that the mode identified by them is the driver of disorder in the perovskite lattice? What is the evidence of the same?*

Response:

We wish to thank the reviewer for revealing the ambiguity regarding our definition of the killer mode. We did not mean to infer any link between the killer mode and thermal disorder in the perovskite system, nor would we have evidence to support such a claim. We have added clarifications on what we consider to be a killer mode for the purposes of this paper, and we believe that this has improved the text significantly (List of Changes #1).

List of Changes

1. PREVIOUS VERSION (lines 27-31)

The overall strength of carrier-phonon coupling for various phonon modes is described by the experimentally-accessible Huang-Rhys factor, S . If the Huang-Rhys factor for one mode is much higher than all the others, the mode is sometimes referred to as a ‘killer mode’, as it becomes predominantly responsible for carrier scattering, effectively ‘killing’ performance in many applications.

NEW VERSION (lines 57-64)

We note that the relative coupling strengths for different LO phonon modes are linked to the magnitude of the macroscopic electric field that each mode generates. Furthermore, if the relative contribution of one mode (LO or otherwise) dominates over those of all other phonons, the mode in question may be referred to as a ‘killer mode’. This term was introduced for molecular semiconductors, where the phonon mode that contributes most strongly to electron-phonon coupling also drives thermal disorder. However, we do not consider thermal disorder here, instead defining a killer mode as a mode that dominates carrier-phonon scattering in a certain material.

Comment 2. *The authors should elucidate better why $S \sim 0.4$ is not consistent with large polaron model in the main manuscript. This is the crucial component of the manuscript and the discussion is mostly relegated to the SI, with most of the discussion there on self-trapped excitons. I understand that the large Huang-Rhys factor does indicate strong coupling to phonons, but why should it lead to the seven-fold enhancement in the effective mass? This is not clear from any of the equations presented in Section S5 of the SI. An important reference in this context is the work from Mathies and co-workers (Nature Communications, 9, 2525(2018)), where they theoretically estimated the displacement of PES and experimentally measured it. They suggest that this is intrinsic characteristic of metal halide perovskites mainly dictated by the inorganic lattice component. Similar observations were made by Batignani et al in Nature Communications, 9, 1971 (2018).*

Response:

We kindly thank the reviewer for this remark. The estimation of the intrinsic Huang-Rhys factor is indeed a key point to our discussion, and should be presented in a clear and precise manner. Regarding precision, we note that in the previous version of the text we made a conservative estimate for the intrinsic value of the Huang-Rhys factor (using equations 4 and 5 in the SI), by treating the polaron mass as a variable parameter, and considering a lattice temperature of 10 K. However, the temperature at which our MPRS experiment was conducted was in fact lower, at 3.8 K. After considering the comment of the reviewer, we have adjusted the estimate to better reflect our experiment. As such, we now report that the required polaron mass that would explain an intrinsic Huang-Rhys factor $S=0.39$ is in fact over 20 times larger than the bare electron mass. Note that this estimate corresponds to a free-electron-polaron, as opposed to an excitonic polaron, where an even greater mass enhancement would be implied by an intrinsic $S=0.39$. Conversely, the Fröhlich constant of CsPbBr_3 , implies a much smaller mass enhancement, with a reported value of around 50%, and therefore a 20-fold enhancement is unphysical, meaning that the experimental value of the Huang-Rhys factor cannot be explained intrinsically.

Regarding the clarity of the discussion, we note that our estimate follows the argument developed in this field by J.J. Hopfield, in his seminal 1959 paper on the topic of edge emission (J. Phys. Chem. Solids, 10, 110-119, (1959)), which has been referenced accordingly. The argument of Hopfield represents the norm in the field when discussing the uncharacteristically high Huang-Rhys factors of various materials that may instead be explained by a localization-related enhancement of the Fröhlich interaction. To better illustrate this argument, we note that in equations 4 and 5 of the SI, the temperature and polaron mass both enter the expression for the maximum wave vector k_0 in the case of free polarons. This latter quantity is the cutoff value for the summation of eq. 5 which determines the Huang Rhys factor S . As such, a given intrinsic value of S at a fixed temperature requires a certain polaronic mass enhancement. We thus determine that the relatively low mass enhancement in CsPbBr_3 cannot account for the recorded value of S , and surmise that extrinsic factors must have played a role in enhancing the carrier-lattice interaction. This is explained more clearly in the current version of the manuscript, thereby elaborating on how section S5 of the SI is related to the main text. (List of Changes #2)

Finally, we kindly thank the reviewer for the additional references. They indeed address a similar topic to ours, i.e. the phonon modes involved in the polaron formation process, while maintaining a few key differences. We believe that the addition of these references to our bibliography has significantly improved our narrative. (List of Changes #3)

List of Changes

2. PREVIOUS VERSION (lines 158-167)

The Fröhlich constant also determines the intrinsic value of the Huang-Rhys factor (Supplementary Information, Section 5). However, considering free polarons, the value of $S \sim 0.4$ requires a seven-fold enhancement in the carrier effective mass, which would require $\alpha \gg 1$.^{43,45,46} The value of $\alpha \approx 2$ for CsPbBr_3 corresponds to an intermediate carrier-lattice coupling regime,⁹ where the effective mass correction remains relatively small. This fact is corroborated by a comparison between cyclotron resonance measurements and estimations of the bare carrier mass via density functional theory, which suggest that an effective mass enhancement of less

than $\sim 50\%$ is more realistic.^{33,47,48} Therefore the experimentally-obtained S factor cannot be intrinsic for CsPbBr_3 .

NEW VERSION (lines 267-280)

We further note that the Huang-Rhys factor is directly proportional to the Fröhlich constant (Supplementary Information, Section 6). Additionally, the magnitude of the Huang-Rhys factor is determined by a summation over all available phonon wave-vectors.⁶⁰ In the case of free electron or hole polarons, the phonon occupation is determined by the thermal energy and polaron mass (Supplementary Information, equations 4 and 5). At the 3.8 K corresponding to our MPRS experiment, to reach the value $S = 0.39$, the low thermal energy must be compensated by an absurdly large polaron mass, more than 20 times the bare electron mass. This would require $\alpha \gg 1$.^{57,59,61} For an excitonic polaron, an even greater mass enhancement is likely necessary, as suggested above. For CsPbBr_3 , the value of $\alpha \approx 2$ corresponds to an intermediate carrier-lattice coupling regime, where the effective mass correction remains relatively small. Cyclotron resonance measurements and estimations of the bare carrier mass via density functional theory suggest that the polaron is only $\sim 50\%$ heavier,^{18,48,62,63} which is insufficient to account for the measured Huang-Rhys factor, instead resulting in $S \approx 0.07$.

3. PREVIOUS VERSION

Papers not referenced

NEW VERSION (lines 75-82)

It has been suggested in two separate studies that polaron formation in hybrid perovskites involves mainly low-frequency phonons related to the inorganic sublattice.^{26,27} However, the investigated materials had a tetragonal crystal structure, as opposed to the orthorhombic structure of CsPbBr_3 , such that the relevance of these results in the inorganic species is not immediately apparent. Moreover, both studies employed the technique of femtosecond impulsive stimulated Raman spectroscopy which, once again, cannot directly probe the LO phonons that contribute to the Fröhlich interaction in centro-symmetric materials.²⁸

Comment 3. *Let us consider that the mode M_0 is indicative of carrier localization at defects, meaning it is the mode along which the lattice re-organizes in the presence of trapped charges in the lattice. Why should it have any relevance in free carrier transport or hot carrier thermalization. In the case of transport, does M_0 mode play a role in de-trapping mechanism? There is however no evidence of any such de-trapping processes or a role of phonons, at least to my knowledge.*

Response:

This comment is highly appreciated, as it has revealed a place of improvement for the clarity of our manuscript. We have modified the text to more precisely reflect the intended message, and now the significance of mode M_0 is presented throughout the manuscript. As such, the significance of a killer phonon mode is now properly highlighted (List of Changes #4), and we more clearly explain why M_0 is in fact a killer mode (List of Changes #5).

Most significantly, we emphasize that, as the dominant mode with regards to the Fröhlich interaction, M_0 is the mode along which the lattice reorganizes, regardless of the type of carrier, or of its trapping state. This is because the macroscopic electric field of LO phonons participating in the Fröhlich interaction cannot be modulated by microscopic local factors. This means that the strong *relative* contribution of M_0 compared to other LO modes cannot be a consequence of localization, or of the type of carrier involved in the Fröhlich interaction. We additionally remark that M_0 is in fact a normal mode of the perovskite lattice and therefore, contrary to the reviewer's assessment, is not indicative of extrinsic factors such as carrier localization at defects, but instead reflects the intrinsic properties of the CsPbBr_3 lattice (List of Changes #6).

The dominant contribution of M_0 to the Fröhlich interaction should therefore be sufficient evidence of the involvement of this LO phonon mode in carrier transport and thermalization.

List of Changes

4. PREVIOUS VERSION

No illustration of the impact of a killer mode on carrier-phonon coupling.

NEW VERSION (lines 64-67)

To illustrate the significance of killer mode identification, it is worth noting that, in the context of orthorhombic perovskites like CsPbBr_3 , the presence of such a mode would lead to great simplifications in the description of carrier-phonon coupling, reducing the number of significant contributions to a single one, from a total 60 normal modes.¹⁷

5. PREVIOUS VERSION (lines 72-80)

The main observation from the MPRS experiment is that the LO mode M_0 and its overtones dominate the resonant Raman spectrum. In the discussion section we will emphasize how the observation of an LO phonon progression is only possible under resonant scattering conditions. Note that the CsPbX_3 lattice also supports LO modes at energies below M_0 .^{16,27,29} These modes lie close to the non-polar TO modes M_1 , M_2 , and M_3 , but are not distinguishable in the measured spectrum. Regardless, the lack of multiphonon scattering for LO phonon modes other than M_0 means that the strength of carrier-phonon coupling involving M_0 far exceeds that of all other Fröhlich-active modes, and therefore M_0 represents the killer mode in CsPbBr_3 .

NEW VERSION (lines 163-172)

Before proceeding we must note that the CsPbX_3 lattice also supports LO modes at energies below M_0 .^{30,43,44} These additional modes should be energetically close to the non-polar TO modes M_1 , M_2 , and M_3 , and may therefore be hidden in the measured spectrum. However, strong Fröhlich scattering comparable to that involving mode M_0 should still produce visible overtones of these hidden modes. In the MPRS spectrum, no multiphonon progressions are observed in the low-energy range, meaning that M_0 must be the dominant contributor to the Fröhlich interaction in CsPbBr_3 . Furthermore, since scattering by M_0 clearly dominates the MPRS spectrum, which contains contributions from both Raman-allowed and forbidden modes, M_0 in fact represents the strongest overall contributor to carrier-phonon scattering (Fröhlich or otherwise) in CsPbBr_3 , and thus behaves as a killer phonon mode in the material.

6. PREVIOUS VERSION (lines 187-189)

We note that the nature of mode M_0 is defined by the LO-TO splitting that occurs in the polar perovskite lattice.^{27,55} As such, M_0 is well-defined as an LO mode throughout the perovskite family, regardless of crystal phase and composition.

NEW VERSION (lines 308-316)

It must however be emphasized that M_0 is in fact a normal mode of the orthorhombic perovskite lattice, meaning it does not arise due to any distortion generated by defects, but occurs naturally in the pristine material as well. Furthermore, exciton localization modifies only the carrier contribution to the Fröhlich carrier-lattice interaction (Supplementary Information, Section 6), meaning that the macroscopic electric field of each LO phonon remains unaffected. That means that a free carrier (or, indeed, any other type of carrier) will also interact most strongly with mode M_0 , out of all available LO modes. The dominance of M_0 over the Fröhlich interaction is thus not exclusively defect-related, instead reflecting the intrinsic properties of the CsPbBr₃ system.

Comment 4. *With such large S, one should be able to observe vibronic replicas in the linear absorption and PL spectra, is it the case? Of course, if this is related to self-trapped exciton, then one might expect to see the replicas exclusively in the emission spectra (at low temperatures).*

Response:

The reviewer is, of course, correct. Phonon replicas can indeed appear in the low-temperature emission spectra for such values of S. For example, in reference 2, a phonon progression was observed in the low-temperature PL spectra of the same type of CsPbBr₃ NCs, with the same Huang-Rhys factor. However, the experimental conditions for the PL measurement reported in the current study are different from the ones of reference 2, such that we do not necessarily expect to see phonon replicas in our PL data. This fact is emphasized in the current version of the text (List of Changes #7)

We note however, for the purposes of this discussion, that the Huang-Rhys factor $S=0.39$ falls within the typical range for CsPbBr₃ NCs. Additionally, M_0 is a well-known LO phonon mode of the perovskite lattice, which is observed strictly due to our resonant scattering arrangement, as it is normally Raman-forbidden. As such, M_0 can be convincingly linked with the Fröhlich interaction in CsPbBr₃ by comparing our measurement with typical non-resonant Raman experiments. We therefore submit that the MPRS spectrum and its interpretation in the text constitute sufficient evidence of the strong Fröhlich interaction in CsPbBr₃.

List of Changes

7. PREVIOUS VERSION

Absence of phonon replicas in PL spectrum not addressed.

NEW VERSION (lines 245-251)

Finally, we note that one previous study has shown that phonon replicas may appear in the low temperature PL spectrum of CsPbBr₃ NCs with a similar Huang-Rhys factor.² We remark however that the linewidth of our PL emission (Supplementary Information, Figure S1) is twice as large as that in the respective reference. Since phonon replicas are only visible if the linewidth is

sufficiently low in comparison to the phonon energy, a doubling of the linewidth can thus completely erase any trace of multiple PL bands, even for much higher S values than the one found here.⁵⁶

Comment 5. *The authors also claim that the THz response, which is indicative of dielectric relaxation, is also defect mediated. Firstly, it has been suggested and partly proven that CsPbBr₃ NCs are almost defect free, as indicated by their high PL quantum yields. Even in the presence of defects, which may be of the order of 10^{14-15} cm⁻³, which is one in a hundred thousand unitcells, in this case less than 0.1 defects per nanocrystal. Why should such small density of defects dictate the dielectric relaxation? I agree that the slow picosecond dielectric relaxation timescales are striking, but I am not convinced that this is a result of defects. Is there a role played by the organic ligands that cap the nanocrystal?*

Response:

The reviewer has raised an interesting point regarding defects in perovskites in general, as well as in the particular case of perovskite NCs. The text has been modified to reflect the fact that the most prevalent defects in perovskite NCs do not actually lower the PL quantum yield (List of Changes #8). Furthermore, defects may in fact be more common in NCs than the estimation of the reviewer, if one considers that the ligand shell does not perfectly passivate the NC surface (De Roo et al, *ACS Nano* 2016, 10, 2, 2071–2081).

Regarding the ligand contribution, we note that the THz-TDS analysis considers the volume fraction occupied by the CsPbBr₃ NCs in the solution, specifically excluding the organic shell (Section 8 of the SI). Noting that the refractive indexes of the capping molecules (oleic acid and oleylamine), as well as that of the solvent (hexane) are comparable to each other, their contributions are simultaneously subtracted from the measured signal via the Maxwell-Garnett equation. As such, the THz response in Figure 2 corresponds strictly to CsPbBr₃, and the low-frequency relaxation is likely unrelated to the ligands. This fact is now clearly stated in the text. (List of Changes #9)

List of Changes

8. PREVIOUS VERSION

No mention of defect tolerance of perovskite NCs.

NEW VERSION (lines 335-339)

Note that most defects in the perovskite lattice have a favorable energetic position close to the edges of the valence and conduction bands,⁷² and do not increase the rate of nonradiative recombination. Therefore the proposition that defects may contribute to the observed dielectric response does not contradict the excellent PL efficiency of CsPbBr₃ NCs.

9. PREVIOUS VERSION

No direct mention of the fact that ligands are excluded from the THz-TDS analysis.

NEW VERSION (lines 346-350)

We further note that in the current scenario, the analysis was conducted in such a way as to extract only the response of the perovskite lattice to the applied field (Supplementary information, section 8). That is, the organic ligands were excluded from the volume fraction of the NC inclusions. Consequently, the dipolar relaxation is not attributable to the ligands.

Reviewer 2

Comment 1. *Failing of the Fröhlich model: the authors entirely neglect the possibility that the Fröhlich model is failing here. To see why this is relevant, consider that the Fröhlich model describes the interaction of one electron with one LO phonon. Thus, the model is based on the harmonic approximation which is not valid for HaPs at the experimental conditions of this work (see, e.g., Yaffe et al. PRL 118, 136001 (2017), Gold-Parker et al. PNAS 115, 11905–11910 (2018), Gehrman and Egger Nature Communications 10, 3141 (2019), Lahnsteiner and Bokdam arXiv:2101.06099 [cond-mat.mtrl-sci] (2021)). If the assumption is invalid, why should the model work? Isn't it at all likely the mode is incorrect and HR still intrinsic? This should be discussed.*

Response:

We kindly thank the reviewer for this observation. We are of course familiar with the work of Yaffe et al, since it is part of our bibliography. Note however that the experiments therein (as well as the other references) clearly show that anharmonicity effects do not play a role in the perovskite lattice at cryogenic temperatures. Since our MPRS experiment was carried out at 3.8 K, the Fröhlich model is, in fact, valid. In response to this comment, we have added the suggested references, while also adapting the text to alleviate the confusion regarding the experimental temperature (Additional Changes #1). Finally, we now specifically discuss anharmonicity effects in our introduction (List of Changes #10), and later state the fact that the lattice anharmonicity does not play a role in our experiment. (List of Changes #11)

List of Changes

10. PREVIOUS VERSION

No consideration of anharmonicity effects

NEW VERSION (lines 46-56)

We note, however, that anharmonic lattice vibrations may account for discrepancies between first-principles calculations within the Fröhlich model and experimentally-measured carrier mobilities.¹⁴⁻¹⁷ Nevertheless, we maintain that the Fröhlich interaction remains a driving mechanism in perovskite physics, which requires proper consideration. Indeed, it has been shown experimentally that the Fröhlich interaction shapes the room-temperature electronic band structure of CsPbBr₃,¹⁸ and is therefore essential in the description of carrier behavior in potential devices. A good starting point for directly probing the Fröhlich interaction, in the absence of anharmonicity effects, is to conduct measurements at cryogenic temperatures, noting that the

impact of the Fröhlich interaction on carrier behavior should only increase towards higher temperatures, as LO phonons become thermally available.

11. PREVIOUS VERSION

No consideration of anharmonicity effects

NEW VERSION (lines 284-290)

Finally, we recall from the introduction that anharmonicity effects may influence carrier-phonon scattering in lead halide perovskites.¹⁴⁻¹⁷ However, these contributions are strictly relegated to elevated temperatures, while being completely suppressed already at 77 K.¹⁴ Since this limit lies well above our experimental temperature of 3.8 K, we can therefore surmise that the experimentally-obtained S factor does not contain any anharmonic contributions, and thus cannot be intrinsically accounted for.

Comment 2. *Interpretation as defect localization: for easy reading the value for the Huang-Rhys factor obtained from the Fröhlich value should be given in the text, providing whether the Fröhlich model, using the parameters provided in the manuscript, underestimates or overestimates the measured Huang-Rhys factor. Also, it will be interesting to compare the obtained HR value to the literature reporting S for defect transitions via multiphonon recombination within the Fröhlich model, e.g., by, for the closely related hybrid HaPs. There, using similar physical models S is MUCH larger ($S \sim 10$). How can this be reconciled with the notion and interpretations presented herein?*

Response:

We thank the reviewer for these suggestions. We have modified the text to clearly show the difference between the expected intrinsic value of the Huang-Rhys factor and the experimental one. The respective discussion has also been modified, and we believe this has improved this section of the text considerably. (List of Changes #2 – repeated below)

Next we would like to address the reviewer's remark about comparing our results with those on hybrid lead halide perovskites. We assume that the reviewer is referring to the work of Kirchartz et al (J. Phys. Chem. Lett. 2018, 9, 939-946), where Huang-Rhys factors on the order of 10-20 are indeed reported. We hereby cite the following statement:

“Note that such large S_{HR} values are not unprecedented and not in conflict with recently reported, much smaller values for similar perovskite systems either, as these experiments studied band-to-band transitions. We, however, are focusing on transitions between delocalized and localized states; the latter typically involve larger structural changes and larger values of S_{HR} .”

The large values of the Huang-Rhys factor in their work are thus reported for “nonradiative recombination via the emission of multiple phonons”, in the presence of mid-gap defect states. This represents an entirely different recombination mechanism from our shallow-trap-assisted, band-to-band, radiative recombination, meaning that their result does not need to be reconciled with ours. To our knowledge, the work of Kirchartz is the only one citing such large values of S . Due to the fundamental differences mentioned above, we believe that including a comparison between our work and theirs may in fact produce unnecessary confusion and have opted not to address this topic in the text.

List of Changes

2. PREVIOUS VERSION (lines 158-167)

The Fröhlich constant also determines the intrinsic value of the Huang-Rhys factor (Supplementary Information, Section 5). However, considering free polarons, the value of $S \sim 0.4$ requires a seven-fold enhancement in the carrier effective mass, which would require $\alpha \gg 1$.^{43,45,46} The value of $\alpha \approx 2$ for CsPbBr₃ corresponds to an intermediate carrier-lattice coupling regime,⁹ where the effective mass correction remains relatively small. This fact is corroborated by a comparison between cyclotron resonance measurements and estimations of the bare carrier mass via density functional theory, which suggest that an effective mass enhancement of less than $\sim 50\%$ is more realistic.^{33,47,48} Therefore the experimentally-obtained S factor cannot be intrinsic for CsPbBr₃.

NEW VERSION (lines 267-280)

We further note that the Huang-Rhys factor is directly proportional to the Fröhlich constant (Supplementary Information, Section 6). Additionally, the magnitude of the Huang-Rhys factor is determined by a summation over all available phonon wave-vectors.⁶⁰ In the case of free electron or hole polarons, the phonon occupation is determined by the thermal energy and polaron mass (Supplementary Information, equations 4 and 5). At the 3.8 K corresponding to our MPRS experiment, to reach the value $S = 0.39$, the low thermal energy must be compensated by an absurdly large polaron mass, more than 20 times the bare electron mass. This would require $\alpha \gg 1$.^{57,59,61} For an excitonic polaron, an even greater mass enhancement is likely necessary, as suggested above. For CsPbBr₃, the value of $\alpha \approx 2$ corresponds to an intermediate carrier-lattice coupling regime, where the effective mass correction remains relatively small. Cyclotron resonance measurements and estimations of the bare carrier mass via density functional theory suggest that the polaron is only $\sim 50\%$ heavier,^{18,48,62,63} which is insufficient to account for the measured Huang-Rhys factor, instead resulting in $S \approx 0.07$.

Comment 3. *Discussion regarding dielectric drag: the discussion with regard to dielectric drag etc. is shallow and not insightful. It is entirely unclear to the reviewer what “dielectric drag” even means, are the authors talking about polarons? What has ever been predicted or explained or tested experimentally that is suggesting dielectric drag even exists? Or that it is a meaningful scientific theory (allowing for deductions that bear substance and are testable by experiment). The reviewer is aware of the partially obscure literature in the context of HaPs and “dielectric drag”, but finds this notion entirely confusing, especially when discussed side-by-side with the well-established polaron concept. The reviewer suggests removing or at least mellowing this discussion.*

Response:

We fully agree that the experimental evidence for dielectric drag is rather scarce and that the concept itself is not intuitive. We are therefore happy to comply with the reviewer’s suggestions. In the interest of avoiding confusion, we have decided to abandon the notion completely, preferring instead to focus only on the possible underlying cause of the low-frequency dielectric response. We have thus removed the term ‘dielectric drag’ from the article. (List of Changes #12, #13, and #14)

List of Changes

12. PREVIOUS VERSION (lines 42-44)

Finally, we argue that dipoles related to defect-neighboring ions may generate a form of ‘dielectric drag’, which may ultimately reduce polaron mobility in inorganic perovskites.

NEW VERSION (lines 118-121)

Finally, we discuss the unexpected low-frequency dielectric response of CsPbBr₃ at room temperature. We tentatively explain the observation as a universal dielectric relaxation linked to the reorientation of dynamic dipoles which may stem from surface defects.

13. PREVIOUS VERSION (lines 209-242)

A similar relaxation has been measured in the hybrid perovskite MAPbBr₃, and it has been attributed to the rotation of MA⁺ cations, which possess an intrinsic dipole moment.^{6,7,62} This response induces screening of the carrier motion in the form of ‘dielectric drag’. Dielectric drag consists of screening the polaronic phonon cloud, thereby reducing the effective mass enhancement of carriers from polaronic effects. Since lighter carriers are more susceptible to scattering,⁴ dielectric drag ultimately reduces carrier mobilities in a material, and may explain the discrepancy between the expected polaronic mobility and the experimentally measured values in lead halide perovskites.⁶ It has been¹¹ suggested that a similar response should be observable in the fully inorganic CsPbBr₃, even in the absence of rotating dipoles, arising instead from the rattling motion of the Cs⁺ cations in their atomic lattice positions.⁶ Indeed, collective displacements of lattice ions in many ionic materials lead to so-called ‘configurational tunneling’, which produces what is called a ‘universal’ dielectric response below the lattice frequency.^{30,31} However, recent results on the dielectric response of single crystals⁶³ suggest that dielectric drag is completely absent in fully-inorganic halide perovskites, while our measurements clearly show that such a response is in fact present in CsPbBr₃ NCs. The absence of the relaxation in bulk single crystals indicates that the dynamic dipoles responsible for the dielectric relaxation cannot be related to the Cs⁺ ion motion. We therefore propose that dielectric drag in CsPbBr₃ is produced by configurational tunneling involving a different entity. The response may be associated with defects, related to the NC surface, but which may also form at grain interfaces in polycrystalline thin films. The link between dielectric drag and lattice defects in CsPbBr₃ is supported by molecular dynamics simulations, which have shown that the positions of lattice ions at halide vacancies fluctuate on a similar time scale (5 ps)⁶⁵ as our measured dielectric relaxation ($\tau = 32$ ps) (by comparison, relaxation due to Cs⁺ motion occurs on a time scale of around 100 fs²⁶). The charge density around a defect depends strongly on the positions of neighboring ions, such that these fluctuations may induce a dipole which can respond to an electromagnetic field, as required for configurational tunneling. Finally, we note that the possible involvement of defects in the enhancement of the Fröhlich interaction as well as the generation of dielectric drag does not imply that the two effects are related in any way. This is immediately apparent when considering that the first mechanism in fact describes the effects of localization on the carrier-phonon interaction, while the defect itself may be substituted by other symmetry breaking factors, whereas the second mechanism is strictly related to the dynamics of defect-neighboring ions, with no relation to phonons or localized carriers.

NEW VERSION (lines 322-356)

A similar relaxation has been measured in the hybrid perovskite MAPbBr₃, and it has been attributed to the rotation of MA⁺ cations, which possess an intrinsic dipole moment.^{6,7,70} It has been suggested that a similar response should be observable in the fully inorganic CsPbBr₃, even in the absence of rotating dipoles, the response arising instead from the rattling motion of the Cs⁺ cations in their atomic lattice positions.⁶ Indeed, collective displacements of lattice ions in many ionic materials lead to so-called ‘configurational tunneling’, which produces what has been referred to as a ‘universal’ dielectric response below the lattice frequency.^{45,46} However, recent results on the dielectric response of single crystals⁷¹ suggest that such a relaxation is completely absent in fully-inorganic halide perovskites, while our measurements clearly show that a low-frequency response is in fact present in CsPbBr₃ NCs. The absence of the relaxation in bulk single crystals indicates that this response cannot be related to the Cs⁺ ion motion. We therefore propose that the low-frequency relaxation we observed is produced by configurational tunneling involving a different entity. The response may, for example, be associated with defects, related to the NC surface. Note that most defects in the perovskite lattice have a favorable energetic position close to the edges of the valence and conduction bands,⁷² and do not increase the rate of nonradiative recombination. Therefore the proposition that defects may contribute to the observed dielectric response does not contradict the excellent PL efficiency of CsPbBr₃ NCs. The link between the dipolar relaxation and lattice defects in CsPbBr₃ is supported by molecular dynamics simulations, which have shown that the positions of lattice ions at halide vacancies fluctuate on a similar time scale (5 ps)⁷³ as our measured relaxation ($\tau=32$ ps) (by comparison, relaxation due to Cs⁺ motion occurs on a time scale of around 100 fs¹⁴). The charge density around a defect depends strongly on the positions of neighboring ions, such that these fluctuations may induce a dipole which can respond to an electromagnetic field, as required for configurational tunneling. We further note that in the current scenario, the analysis was conducted in such a way as to extract only the response of the perovskite lattice to the applied field (Supplementary Information, section 8). That is, the organic ligands were excluded from the volume fraction of the NC inclusions. Consequently, the dipolar relaxation is not attributable to the ligands. Finally, we note that the possible involvement of defects in the enhancement of the Fröhlich interaction as well as the generation of a dielectric response does not imply that the two effects are inter-related. This is immediately apparent when considering that the first mechanism describes the effects of localization on the carrier-phonon interaction, while the defect itself may be substituted by other symmetry breaking factors. Conversely, the second mechanism is strictly related to the dynamics of defect-neighboring ions, and not the carriers themselves.

14. PREVIOUS VERSION (lines 253-255)

We suggest that in the absence of freely-rotating dipolar cations, the strong relaxation observed in the fully-inorganic CsPbBr₃ is instead linked to defect-related dynamic dipoles, leading to a distinct form of dielectric drag.

NEW VERSION (lines 387-395)

Additionally, the low-frequency dielectric response of CsPbBr₃ merits consideration. Here we tentatively linked the observed Debye-like dielectric relaxation to surface defects, due to the low probability of finding internal imperfections under a NC geometry. Presumably, a similar

response can arise from a wide range of defects occurring in the CsPbBr₃ lattice. Consequently, defect-prone materials such as polycrystalline thin films may present a similar dielectric relaxation, which would have an impact on carrier transport, by modifying the amount of carrier screening from defects. A more focused analysis should be conducted to determine whether such a relaxation does in fact occur in other systems, whether its effect on carrier transport is considerable, and whether this effect is positive or negative.

Comment 4. *Discussion about polaron mobility: the present article is alluding quite heavily to polaron mobility. This is a stretch, since the authors do not present any data or analysis that suggests this has even been studied here. Of course it is fine to mention it but perhaps not in such a prominent manner.*

Response:

As with the previous comment, we highly appreciate and agree with this input from the reviewer. In response to this we have also removed the notion of polaron mobility from the manuscript, again avoiding any overly-speculative statements (List of Changes #15 and #12 – repeated below).

List of Changes

15. PREVIOUS VERSION (lines 12-13)

Our experiments also unearthed a dipole-related dielectric relaxation mechanism which has an impact on polaron mobility

NEW VERSION (lines 12-13)

Our experiments also unearthed a dipole-related dielectric relaxation mechanism which may impact transport properties

12. PREVIOUS VERSION (lines 42-44)

Finally, we argue that dipoles related to defect-neighboring ions may generate a form of ‘dielectric drag’, which may ultimately reduce polaron mobility in inorganic perovskites.

NEW VERSION (lines 118-121)

Finally, we discuss the unexpected presence of a low-frequency dielectric response of CsPbBr₃ at room temperature. We tentatively explain the observation as a universal dielectric relaxation linked to the reorientation of dynamic dipoles which may stem from surface defects.

We thank the reviewer of urging us, with comments 3 and 4, to be careful with simplified and possibly disputable pictures of physical reality.

Reviewer 3

Comment 1. *The Results section starts by introducing the acronym 'NCs' for nanocrystals, without defining it. In itself this is of course only a minor stylistic error, but it underlines the fact that no reference is made to the study being about nanocrystals in the introduction. Lead halide perovskites have indeed achieved remarkable success in recent years in many applications, but their standout achievements have been in the bulk form in thin-film solar cells. Nanocrystal behaviour is not always the same as bulk behaviour (see e.g ten Brinck et al. ACS Energy Lett. 2019, 4, 2739–2747, Akkerman et al Nat. Mater 2018, 17, 394-405) though of course there will be considerable overlap. The introduction and later part of the paper should therefore be modified to clarify that the study was performed on nanocrystals, and to justify whether or not the results can be applied to and compared to bulk CsPbBr₃.*

Response:

Following the reviewer's suggestion, the introduction has been modified to note that the results pertaining to the Fröhlich interaction in CsPbBr₃ NCs are also valid for the bulk system. This fact is now explained in detail in the added Section 5 of the Supplementary Information. As such, we believe that we now provide a clear illustration of the significance of our work to CsPbBr₃ and other lead halide perovskites, also outside of the NC system (List of Changes #16). We hereby thank the reviewer for this very constructive comment.

List of Changes

16. PREVIOUS VERSION

No comparison between NC and bulk system.

NEW VERSION (lines 33-39)

Here, we report on the Fröhlich interaction in CsPbBr₃ nanocrystals (NCs) at cryogenic temperatures, identifying a possible reason for the strong exciton-phonon coupling that is observed. We further argue that the case of NCs is representative also for bulk systems and, moreover, that there should be no significant structural differences between the CsPbBr₃ lattices at cryogenic and room temperature (Supplementary Information, Section 5). Consequently, our analysis provides a general view of the Fröhlich interaction in CsPbBr₃ and related materials.

Comment 2. *On a related note, Raman measurements on single crystals are used to justify the attribution of certain Raman peaks to TO phonons on line 66, but the Yaffe et al. paper cited does not give much attention to phonons except in first-principles DFT calculations in the supplementary material. This argument thus needs either more clarification, or strengthening from other sources.*

Response:

We thank the reviewer for this observation, which has prompted important clarifications in our main text. The discussion regarding the TO mode symmetry attribution has hereby been modified to reflect the main reference that was used (Calistru et al, J. Appl. Phys., 1997, 82, 5391). We have also added more details to our reasoning on the subject, in order to avoid further confusion (List of Changes #17).

We further remark that, while the symmetry of the low energy modes might remain a topic of discussion, the main LO phonon mode is unambiguously M_0 , as evidenced by the MPRS spectrum, which in our case contains both Raman-allowed (non-polar) and Raman-forbidden (polar, i.e. LO) phonon modes. Since M_0 is a well-known normal mode of the perovskite lattice, its dominance over the MPRS spectrum of CsPbBr_3 is sufficient for the purposes of this paper, namely to establish its status as a killer mode, which has the most significant contribution to carrier-phonon scattering and related effects.

List of Changes

17. PREVIOUS VERSION (lines 65-71)

We attribute these peaks to TO phonons, as reported in Raman measurements of single crystals.²⁶ This assessment is additionally supported by an analogy with a known vibrational spectrum of an orthorhombic perovskite such as CsPbCl_3 , where mode energies and splittings are renormalized according to the different anion mass and octahedral rotation angle.²⁷

NEW VERSION (lines 149-159)

For a tentative symmetry assignment of the low-energy modes, we compare the Raman spectrum of CsPbBr_3 with the known vibrational spectrum of the orthorhombic perovskite CsPbCl_3 .⁴³ When the two materials have the same crystal phase, the phonon modes of CsPbCl_3 all have equivalent modes in CsPbBr_3 . The mode energies and splittings are renormalized according to the differences in ionic mass and octahedral rotation angles, respectively. The renormalized mode energies can be obtained by modeling the phonons as harmonic vibrations of chains of alternating lead and halogen atoms. While the proposed model is quite simple, we find very good agreement between the CsPbCl_3 phonon energies adjusted with respect to a change in anion from Cl to Br and the energies of the CsPbBr_3 peaks observed in the spectrum of Fig. 1a. We thus tentatively identify M1, M2, and M3 as TO modes for CsPbBr_3 .

Comment 3. *Other points at which additional clarity (or stylistic improvement) is needed in the results section include:*

-line 47: at which low temperatures do CsPbBr_3 NCs have an orthorhombic crystal structure?

In general, the temperatures at which the measurements were taken need to be made clearer, and the term 'low temperature' better defined at least once in the paper.

-line 49: what are the respective bandgap values?

-Figure 1d: what temperature does this crystal lattice represent? How was it generated, or is it merely an illustration of a generic orthorhombic perovskite lattice? This subfigure does not seem to add to the narrative.

-line 55: A reference to Ramade et al would be a good addition: Appl. Phys. Lett. 112, 072104 (2018);

-Figure 2b: the legend should read 'Calculation' or 'Theory' for the red data, and the 'relative cross-sections' legend in the bottom left should be eliminated as it duplicates the data points in the figure itself

Response:

The above issues have now been addressed. We kindly thank the reviewer for these observations. (Additional Changes #1, #2, #3, #4, #5)

As the reviewer remarks, (former) Figure 1d does not add to the narrative of the text (being only an illustration of the orthorhombic lattice). Consequently, the figure has been removed from the manuscript, so as not to draw focus away from the actual results (Additional Changes #4).

Comment 4. *Line 54: What is the uncertainty in the 19.2 meV shift?*

Response:

Following this comment by the reviewer, we have modified the text to better describe the positions of the consecutive phonon replicas. We note therein that the position of each replica is slightly lower than exact multiples of the LO phonon energy of 19.2 meV. This is taken as a sign of forbidden scattering via the Fröhlich interaction, which is well documented (see for example Cardona, 1982, *Light Scattering in Solids II*). We believe that this further strengthens our argumentation that the observed mode is in fact an LO mode of the perovskite lattice, that can only be observed under resonant conditions such as those of our experiment. (List of Changes #18)

List of Changes

18. PREVIOUS VERSION (lines 52-55)

The MPRS spectrum of CsPbBr₃ NCs in Fig. 2a shows several intense peaks shifted by multiples of 155 cm⁻¹ (19.2 meV). These represent the multiphonon progression of a Fröhlich-active LO phonon mode, labelled M₀, also observed in previous studies.^{2,11,20,21}

NEW VERSION (lines 130-140)

The MPRS spectrum shows several intense peaks shifted by near multiples of 155 cm⁻¹ (19.2 meV). These represent the multiphonon progression of an LO phonon mode, labelled M₀, also observed in previous studies.^{2,12,35-37} In the discussion section it will be shown that the observation of mode M₀ in this study is strictly linked to the resonant experimental conditions in conjunction with symmetry-breaking factors. As is typical of multiphonon Raman scattering via the Fröhlich interaction,³⁸ the overtone energies were slightly lower than exact multiples of the phonon energy, each peak exhibiting a red-shift smaller than their respective FWHM. This is a consequence of the forbidden nature of Fröhlich scattering at the center of the Brillouin zone (see Discussion), such that the observed peaks are the result of scattering by lower-energy LO phonons of non-zero momentum.

Comment 5. *Was it not possible to observe further overtones - a straight line fit to only three data points is not very convincing.*

Response:

We kindly note that the inset to the current Figure 1a demonstrates that the FWHM of the overtones increases with increasing order. This constitutes sufficient evidence that the main scattering mechanism is resonant Raman scattering (RRS), as opposed to hot carrier luminescence (HCL), where the FWHM would instead decrease towards higher orders, as explained in the Supplementary Information file. Consequently, the linearity of the dependence is not crucial to this assessment, and we maintain that the data presented is more than convincing. Furthermore, the line is treated only as a guide to the eye and no use is made of any linear fit for these data points in the text.

Additionally, we would like to emphasize that the observation of three consecutive peaks with such intensity is already quite remarkable, given the material parameters. Consequently, the relevance of these data points cannot be overstated. We further note that, while there was in fact an indication of a fourth peak, it was difficult to draw any conclusions regarding its width and intensity, due to the low signal-to-noise ratio in its spectral range (this can be seen in the figure source data provided with the current resubmission).

In reply to this reviewer comment, we have corrected the text related to the inset, to highlight the fact that RRS and HL result in different behaviors of the overtone FWHM with increasing scattering order (as detailed in Section 4 of the Supplementary Information). We are hereby grateful to the reviewer for prompting this improvement (List of Changes #19).

List of Changes

19. PREVIOUS VERSION (lines 55-57)

The broadening of the overtones with increasing order (Fig. 2a inset) suggests that the multiphonon spectrum is caused by resonant Raman scattering (Supplementary Information, Section 4).

NEW VERSION (lines 140-142, additional changes in Supplementary Information, Section 4)

The broadening of the overtones with increasing order (Fig. 1a inset) suggests that the multiphonon spectrum is produced by resonant Raman scattering, as opposed to hot luminescence (Supplementary Information, Section 4).

Comment 6. Line 78: what is the nature of the 'carriers' being considered? Do they mean 'charge-carriers'? In which case, are they excitons or free-carriers?

Response:

We kindly thank the reviewer for revealing the confusion surrounding this term. The introduction has been modified to clarify this topic, noting that 'carriers' in CsPbBr₃ are mainly excitons. Consequently, our analysis deals, strictly-speaking, with exciton-phonon coupling (List of Changes #20). We also note, however, that the qualitative description of the Fröhlich interaction does not depend on the type of carrier involved (List of Changes #21).

List of Changes

20. PREVIOUS VERSION

No distinction between 'carrier' types with regards to Fröhlich coupling.

NEW VERSION (lines 20-28)

Most discussions on polaron physics in perovskites concern hybrid organic-inorganic lead iodides, from which highly efficient solar cells have been developed. These materials have a low exciton binding energy, enabling conduction with free electrons and holes that strongly polarize the lattice. However, some halide perovskite species, such as the fully-inorganic CsPbBr₃,⁸ or layered two-dimensional hybrid perovskites⁹ exhibit room-temperature stable excitons, which are highly

desirable for light emission applications. Due to the high exciton binding energy, the Fröhlich interaction in these materials will produce excitonic polarons. Consequently, the interaction is expected to be very weak, as the phonon clouds of electron and hole polarons mutually compensate when overlapping.

21. PREVIOUS VERSION

No mention of qualitative equivalence between exciton-phonon and single-carrier-phonon coupling via the Fröhlich interaction.

NEW VERSION (lines 261-266)

While in the case of excitons the reduced mass is sometimes used to calculate the Fröhlich constant, this approach does not provide an accurate picture of the behavior and physics of excitonic polarons. Still, any argument regarding the strength of electron-phonon coupling remains qualitatively valid for exciton-phonon coupling as well, keeping in mind that the latter is likely weaker than the former, due to the compensation of the electron and hole charges upon exciton formation, which diminishes their respective phonon clouds.

Comment 7. *On page 7, the usage of an additional dielectric function data point from a separate study needs additional clarification. Was this value obtained by similar methods? If not, it should be justified why it is appropriate to mesh it with the data in this paper since no one method measures the 'true' value of the dielectric function, and there is often large variation depending on the technique used and form of the material.*

Response:

We fully agree with the reviewer, that one can always find large variations between measurements of the dielectric function. Consequently, the reported values of $\epsilon_S = 20.5$ and $\epsilon_\infty = 4.8$ are viewed only as an estimate. In that sense it is instructive to compare the experimental result with calculations, as done in the text. For example, Yettapu et al. obtained $\epsilon_S = 15.6$ and $\epsilon_\infty = 4.12$, which is reasonably close to our result. As such, our reported values of the dielectric constants are not implausible, and this should suffice for the purposes of this discussion. Since the absolute values of ϵ_S and ϵ_∞ mainly influence our calculation of the Fröhlich constant and the corresponding Huang-Rhys factor, we note that even if we consider an improbably high $\epsilon_S = 40$ and an absurdly low $\epsilon_\infty = 2$, and thus a three times larger dielectric contrast, the Huang-Rhys factor $S=0.39$ still cannot be intrinsically accounted for. This fact is better reflected in the current version of the text. (List of Changes, #18)

List of Changes

22. PREVIOUS VERSION

No comment on the validity of the values for ϵ_S and ϵ_∞ in the context of our work.

NEW VERSION (lines 281-284)

Note that the requirement of an unrealistic carrier mass is unlikely to result from a faulty estimation of ϵ_S and ϵ_∞ , since the values used here are reasonably close to previous reports. Furthermore, the polaron mass enhancement is linearly dependent on $\alpha \propto \bar{\epsilon}^{-1}$ which is a relatively slowly-varying function of the two dielectric constants.

Comment 8. *Additionally, the value of 20.5 in reference 32 seems to have been measured at a different temperature from the measurements in this paper (although as mentioned above, the temperature at which measurements were taken in this paper is conspicuously opaque) - whether this temperature difference matters needs to be discussed.*

Response:

We respectfully note that, in fact, the value of $\epsilon_S = 20.5$ is practically temperature-independent between liquid helium and room temperature in the cited reference. As such, the chosen value is suitable for this analysis, regardless of our experimental conditions. As mentioned above, the previously-omitted experimental temperatures are now clearly specified in the revised manuscript (Additional Changes, #1).

Comment 9. *Finally, the fact that the epsilon_S value obtained from the fit is the same as that inputted from reference 32 needs discussion: it is dubious to note that 'epsilon_S is the same as the experimental value at 108 MHz' as if to strengthen the credibility of the value obtained from the fit, when that very experimental value was inputted into the fit in the first place!*

Response:

We thank the reviewer for identifying this unclear statement. The intent here was to underscore the fact that the added data point fixes the value of the static dielectric constant. The statement has been rephrased in the revised manuscript to better reflect this fact. (List of Changes #23)

List of Changes

23. PREVIOUS VERSION (lines 111-112)

The value of ϵ_∞ is consistent with previous calculations^{3,18,33} and experimental results,³⁴ while ϵ_S is the same as the experimental value at 1.08 MHz.³²

NEW VERSION (lines 205-206)

The value of ϵ_∞ is consistent with previous calculations^{3,18,33} and experimental results³⁴, while ϵ_S is fixed by the experimental value at 1.08 MHz.³²

Comment 10. *Secondly, the narrative of the paper is rather lacking. The results themselves constitute good data, from which valid conclusions are drawn. However, the results section is presented as a list of the experiments done, without motivation. For example, on line 51: why are further investigations carried out using these techniques in particular? Figure 1 in general does not add much to the narrative and could be left in the SI without loss to the paper, other than that there would only be one figure left. In the discussion, there should at least be an introductory sentence justifying the upcoming analysis and why it follows from the results, though the discussion itself is convincing.*

Response:

This comment has brought about major corrections in the manuscript. To address the lack of motivation for the experiments, we have highlighted the fact that MPRS is a powerful technique for

characterization of the carrier-phonon interaction in a material, which is capable of analyzing otherwise Raman-forbidden polar-optical-phonon (Fröhlich) scattering in centro-symmetric crystals, thereby providing a complete picture of the phonon spectrum of CsPbBr₃. We also highlight the link between the Fröhlich interaction and the material dielectric function, thus explaining the additional need for a dielectric characterization technique, such as THz-TDS (List of Changes #24).

Regarding Figure 1, we agree it does not add to the narrative and have therefore moved it to the Supplementary Information. (Additional Changes #6)

List of Changes

24. PREVIOUS VERSION

No motivation provided for chosen experimental methods.

NEW VERSION (lines 96-110)

[...] the topic of Fröhlich-active modes in inorganic lead halide perovskites remains open, and requires further analysis via a suitable experimental technique. Addressing this issue represents one of the goals of the current article, in order to provide a clear overview of carrier-phonon scattering in CsPbBr₃. This is achieved via resonant Raman scattering spectroscopy, carried out under conditions that allow the observation of both Raman-active and inactive modes, while also probing a broad range of phonon energies. We further note that the strength of the Fröhlich interaction in a material is directly linked to the polar nature of its crystal lattice. In a highly polar material the Coulomb field of a carrier (or exciton) couples more easily to the polar vibrations (i.e. LO phonons) of the lattice, resulting in strong Fröhlich coupling. The static and high frequency values of the dielectric function provide a means of quantifying the polar nature of a material, such that further insight into the Fröhlich interaction can be gained via dielectric characterization. With that in mind, in this article we used a combination of resonant multiphonon Raman scattering (MPRS) and THz time-domain spectroscopy (THz-TDS - Supplementary Information, Section 8) to describe the carrier-LO phonon interaction in CsPbBr₃ NCs.

Comment 11. *This brings me to my final contention, namely the significance of the paper to the field. Within the study of CsPbBr₃ nanocrystals, it is not entirely clear from the discussion how much of their findings are novel, as opposed to just corroborating previous findings. Perhaps this is just a consequence of the aforementioned weak narrative, however - although their numerical values of S and MO seem to be backed by previous studies, their interpretation of their significance appears novel. Nonetheless, this aspect of the paper needs to be strengthened. Within the field of lead halide perovskites as a whole however, the significance is less clear still. While the examples given in the conclusion of the relevance of LO phonons to the functionality of devices are true, it is less apparent how their findings about the dominant phonon mode in CsPbBr₃ nanocrystals at low temperature are relevant to the behaviour of devices at room temperature. It needs to be clarified how precisely the fact that MO 'effectively controls carrier dynamics in CsPbBr₃' could be of practical use: just because phonons are involved in a process, it does not immediately follow why particular knowledge of them is useful.*

Response:

We consider this to be the most important set of comments regarding our manuscript, and we have therefore gone through substantial effort to consider the point of view of the reviewer and adapt the narrative accordingly. We believe that this has led to a much more impactful manuscript. Due to their generality and interconnectedness, the comments have had an impact on the entirety of the manuscript and as such the specific changes are difficult to address on a point-by-point basis. Nevertheless, the main changes that we would like to point out are listed below.

In order to address the significance of our work to the field, we have expanded the presentation of the status of current research in the introduction portion of our manuscript. It should now become clear that there is a need to investigate the full vibrational spectrum of CsPbBr₃ in order to identify the main contributors to exciton-phonon scattering. We thereby underscore the importance and novelty of our MPRS analysis, in which both Raman-allowed and Raman-forbidden modes are observed, to the field of lead halide perovskites. Additionally, the dielectric function is presented as a crucial parameter in the description of the Fröhlich interaction, underscoring the need for our additional THz-TDS analysis. (List of Changes, #25)

Also in the introduction, we highlight the significance of the Fröhlich interaction as a driving mechanism for the physics of lead halide perovskites. We further note that the impact of the Fröhlich mechanism only increases along with the thermal occupation of LO phonons, thus underscoring its significance in devices operating at room temperature. (List of Changes, #26)

The Conclusions section has also been re-written to better illustrate the implications of our work for the understanding of carrier-phonon interactions in CsPbBr₃ and likely other similar materials. We have shown that the observed discrepancy between the Fröhlich constant and the Huang-Rhys factor not only has implications for the interpretation of the current result, but should be considered in any other related experiments that reveal a Huang-Rhys factor of a similar magnitude (List of Changes, #27). Furthermore, we now provide a clearer account of how our findings impact the current understanding of polaron formation and carrier-LO-phonon interaction in CsPbBr₃. Additionally, we highlight how the observed low-frequency dielectric relaxation may impact transport in the material, noting the necessity of further investigations on this topic. (List of Changes, #28)

Finally, we have stressed the importance of achieving a good fundamental understanding of any material in the interest of potential applications (List of Changes, #29). As such, since the Fröhlich interaction itself represents a critical aspect of perovskite physics, any contribution to its complete understanding is undeniably valuable to the field. We submit that the improved discussion on the implications of our work for polaron physics in perovskites (List of Changes, #28) constitutes a very clear and powerful example of the practical use that our findings may have. The polaron model is, after all, based on knowledge of the energies of involved LO phonons, and as such the disambiguation of this topic is immediately useful in a wide range of theoretical calculations. Furthermore, in our introduction, we have also underscored the impact of killer mode identification on the modelling of carrier-phonon scattering in orthorhombic perovskites (List of Changes, #4 – repeated below). This further highlights the significance of our analysis, as it may effectively reduce the complexity of carrier-lattice interactions by a factor of 60, by emphasizing the dominance of a single phonon mode over all others in CsPbBr₃, and possibly in other similar perovskite species.

List of Changes

25. PREVIOUS VERSION (lines 20-24)

Carrier-LO-phonon coupling via the Fröhlich interaction has been reported in many types of lead halide perovskites.^{2,3,8-14} Recently it has been shown that the Fröhlich interaction may involve various combinations of phonon modes.¹⁵ Moreover, various low-energy LO phonons have been identified as Fröhlich-active,^{15,16} but no overall consensus has been reached regarding the most significant modes involved in the interaction.

NEW VERSION (lines 68-107)

Note that for an accurate experimental overview of carrier-phonon scattering, all modes must be detectable via the chosen technique. As such, conventional Raman scattering and other related methods with similar selection rules are typically insufficient in perovskites, due to the Raman inactivity of LO phonons in centro-symmetric crystals, which excludes the Fröhlich interaction from the scattering analysis. Consequently, while carrier-LO-phonon coupling via the Fröhlich interaction has been reported in many types of lead halide perovskites,^{2,3,11,12,21-25} an experimental investigation of the full phonon spectrum has not been carried out thus far. It has been suggested in two separate studies that polaron formation in hybrid perovskites involves mainly low-frequency phonons related to the inorganic sublattice.^{26,27} However, the investigated materials had a tetragonal crystal structure, as opposed to the orthorhombic structure of CsPbBr₃, such that the relevance of these results in the inorganic species is not immediately apparent. Moreover, both studies employed the technique of femtosecond impulsive stimulated Raman spectroscopy which, once again, cannot directly probe the LO phonons that contribute to the Fröhlich interaction in centro-symmetric materials.²⁸ The contributions of various low-energy LO phonons to the Fröhlich interaction have been reported in lead halide perovskites,^{29,30} but no real consensus exists regarding the modes which dominate the interaction. A somewhat definitive answer was reached for MAPbI₃, where a single effective LO mode around 15 meV was identified as coupling to electronic motion.³¹ Conversely, in CsPbBr₃, two lower-energy modes of the lead-halide sublattice at 3.7 meV and 6.3 meV were seen dominating the resonant Raman scattering spectrum at cryogenic temperatures.³² However, the symmetry of these modes was not well established, and not all of the known normal modes of the perovskite lattice were observed. Moreover, an abnormal decrease of the scattering intensity with increasing temperature introduced additional uncertainty regarding the scattering mechanism responsible for the observed peaks. Further confusion on this topic is generated by the fact that the energies of LO phonons involved in temperature-dependent PL line broadening in CsPbBr₃ do not always coincide with the LO phonon energies found in low-temperature PL spectra.³³ Additionally, the former are much larger than the energies of the two modes mentioned above. Consequently, the topic of Fröhlich-active modes in inorganic lead halide perovskites remains open, and requires further analysis via a suitable experimental technique. Addressing this issue represents one of the goals of the current article, in order to provide a clear overview of carrier-phonon scattering in CsPbBr₃. This is achieved via resonant Raman scattering spectroscopy, carried out under conditions that allow the observation of both Raman-active and inactive modes, while also probing a broad range of phonon energies. We further note that the strength of the Fröhlich interaction in a material is directly linked to the polar nature of its crystal lattice. In a highly polar material the Coulomb field of a carrier (or exciton) couples more easily to the polar vibrations (i.e. LO phonons) of the lattice, resulting in strong Fröhlich coupling. The static and high frequency values of the dielectric function provide a means of quantifying the polar nature

of a material, such that further insight into the Fröhlich interaction can be gained via dielectric characterization.

26. PREVIOUS VERSION

Significance of Fröhlich interaction in perovskites not emphasized. No link revealed between cryogenic and room temperature behavior.

NEW VERSION (lines 340-56)

We remark that the underlying reason for the excellent efficiency of perovskite-based optoelectronic devices is still under debate.¹³ A number of theories exist, including: the scarcity of mid-gap defect states (which would otherwise impact the emission and carrier diffusion lengths through Shockley-Read-Hall recombination), the low rate of nonradiative recombination (determined by the low energies of involved phonons), or the contribution of polaronic carrier screening. Here we address the large polaron model, which appropriately describes carrier-LO-phonon interactions in CsPbBr₃ and related materials. We note, however, that anharmonic lattice vibrations may account for discrepancies between first-principles calculations within the Fröhlich model and experimentally-measured carrier mobilities.¹⁴⁻¹⁷ Nevertheless, we maintain that the Fröhlich interaction remains a driving mechanism in perovskite physics, which requires proper consideration. Indeed, it has been shown experimentally that the Fröhlich interaction shapes the room-temperature electronic band structure of CsPbBr₃,¹⁸ and is therefore essential in the description of carrier behavior in potential devices. A good starting point for directly probing the Fröhlich interaction, in the absence of anharmonicity effects, is to conduct measurements at cryogenic temperatures, noting that the impact of the Fröhlich interaction on carrier behavior should only increase towards higher temperatures, as LO phonons become thermally available.

27. PREVIOUS VERSION (lines 247-251)

The dielectric response of CsPbBr₃, investigated by means of THz-TDS spectroscopy, indicates that the intrinsic strength of the Fröhlich interaction is unable to account for the magnitude of the low-temperature Huang-Rhys factor. We have shown that localization may enhance carrier-phonon coupling sufficiently to explain this experimental observation.

NEW VERSION (lines 358-367)

The low-temperature Huang-Rhys factor $S \approx 0.39$ was determined from the relative resonant Raman cross-section of consecutive overtones. The dielectric response of CsPbBr₃, investigated by means of THz-TDS, indicates that the intrinsic strength of the Fröhlich interaction is incompatible with the low-temperature Huang-Rhys factor. We have shown that localization may enhance carrier-phonon coupling sufficiently to explain this experimental observation. As the Huang-Rhys factor obtained here falls within the typical range of reported values for lead halide perovskites, it thus becomes necessary to consider that such enhancements may in fact play a role in many reported cases of carrier-LO-phonon coupling in this class of materials.

28. PREVIOUS VERSION

Most importantly, our results highlight the influence of M_0 LO phonons on the functionality of lead halide perovskite devices. Through the Fröhlich interaction, M_0 defines the mechanism of fast carrier relaxation, as well as the opposing hot phonon bottleneck and polaron formation mechanisms, which delay carrier cooling.^{57,60} Further, M_0 is the main contributor to carrier-phonon scattering during transport and luminescence processes, but it is also responsible for polaronic effects which protect carriers from scattering. The broad impact in various mechanisms, which may be both favorable and detrimental to device performance, depending on the intended application, serves to highlight the importance of killer mode identification for practical implementation of functional materials. In the current scenario, since transport and relaxation mechanisms are governed by the Fröhlich interaction, M_0 , being by far the most significant Fröhlich-active mode, effectively controls carrier dynamics in CsPbBr₃.

NEW VERSION (lines 368-395)

Concerning the significance of mode M_0 in the context of polaron formation, we recall the recent report that the self-trapping of excitons in CsPbBr₃ NCs involves two lower-energy phonon modes,³² which seems to contradict our results. We note however that the spectrum recorded in the reference was likely incomplete, due to the absence of the resonance with mode M_0 . We hereby underscored the importance of recording the full phonon spectrum in order to properly describe carrier-phonon interactions in any material, as an incomplete analysis can produce misleading conclusions. In the current study we managed to alleviate the existing confusion in the field by recording the full phonon spectrum of CsPbBr₃, including both Raman-allowed and Raman-forbidden modes over a broad energy range. While scattering via M_0 has also been observed, to a certain degree, in other experiments,^{14,36} the forbidden nature of LO phonon scattering resulted in a weak Raman signal and could not provide any indication of the strength of the Fröhlich interaction. By contrast, our experiment was done under resonant conditions, and the symmetry-related selection rules were broken. This fully enabled the observation of 'forbidden' Fröhlich scattering, even in the first order, thereby allowing the quantification of the strength of carrier-lattice interaction. As such, we were able to unambiguously identify M_0 as the main contributor to the Fröhlich interaction, and thereby polaron formation and scattering in CsPbBr₃. This fact should henceforth be considered in the interpretation of scattering-related experiments and polaronic phenomena in this material and likely other perovskite species with an orthorhombic structure. Additionally, the low-frequency dielectric response of CsPbBr₃ merits consideration. Here we tentatively linked the observed Debye-like dielectric relaxation to surface defects, due to the low probability of finding internal imperfections under a NC geometry. Presumably, a similar response can arise from a wide range of defects occurring in the CsPbBr₃ lattice. Consequently, defect-prone materials such as polycrystalline thin films may present a similar dielectric relaxation, which would have an impact on carrier transport, by modifying the amount of carrier screening from defects. A more focused analysis should be conducted to determine whether such a relaxation does in fact occur in other systems, whether its effect on carrier transport is considerable, and whether this effect is positive or negative.

29. PREVIOUS VERSION

No emphasis on significance of fundamental understanding of materials for perovskite device development

NEW VERSION (lines 396-402)

Ultimately, different applications require different material characteristics, and improving the performance of any device is fully reliant on a good understanding of the basic properties of the materials used. For the intended optoelectronic applications of CsPbBr₃, this involves knowledge of carrier relaxation, recombination, and transport, all of which are inextricably linked to the Fröhlich interaction. As such, the defining features of this interaction identified here constitute a step towards the full understanding of CsPbBr₃, and possibly other perovskite species, in the interest of developing novel optoelectronic devices.

4. PREVIOUS VERSION (lines 72-80)

The main observation from the MPRS experiment is that the LO mode M_0 and its overtones dominate the resonant Raman spectrum. In the discussion section we will emphasize how the observation of an LO phonon progression is only possible under resonant scattering conditions. Note that the CsPbX₃ lattice also supports LO modes at energies below M_0 .^{16,27,29} These modes lie close to the non-polar TO modes M_1 , M_2 , and M_3 , but are not distinguishable in the measured spectrum. Regardless, the lack of multiphonon scattering for LO phonon modes other than M_0 means that the strength of carrier-phonon coupling involving M_0 far exceeds that of all other Fröhlich-active modes, and therefore M_0 represents the killer mode in CsPbBr₃.

NEW VERSION (lines 161-170)

Before proceeding we must note that the CsPbX₃ lattice also supports LO modes at energies below M_0 .^{30,43,44} These additional modes should be energetically close to the non-polar TO modes M_1 , M_2 , and M_3 , and may therefore be hidden in the measured spectrum. However, strong Fröhlich scattering comparable to that involving mode M_0 should still produce visible overtones of these hidden modes. In the MPRS spectrum, no multiphonon progressions are observed in the low-energy range, meaning that M_0 must be the dominant contributor to the Fröhlich interaction in CsPbBr₃. Furthermore, since scattering by M_0 clearly dominates the MPRS spectrum, which contains contributions from both Raman-allowed and forbidden modes, M_0 in fact represents the strongest overall contributor to carrier-phonon scattering (Fröhlich or otherwise) in CsPbBr₃, and thus behaves as a killer phonon mode in the material.

Additional Changes

1. Experimental temperature for each measurement now clearly mentioned in the text
2. Reference added for CsPbBr₃ bulk PL energy (Figure S1 of Supplementary Information)
3. Removed Figure 1d
4. Added reference to Ramade et al
5. Replaced 'Calculation' with 'Theory' in Figure 1b (former Figure 2b). Also removed legend at bottom left of the same figure
6. Moved Figure 1 to Supplementary Information
7. Former Figure 2 was separated into new Figures 1(a,b) and 2(a,b), separately showing results for MPRS and THz-TDS analyses, respectively.
8. Various minor changes in phrasing throughout the manuscript

9. Added Section 5 to the Supplementary Information, describing how the NC system is representative for bulk CsPbBr₃.
10. Crystal structure of bulk and NC CsPbBr₃ addressed in Supplementary Information, Section 5
11. Proportionality relation between S and α emphasized in Supplementary Information, Section 6
12. Added statement regarding data availability

REVIEWERS' COMMENTS

Reviewer #1 (Remarks to the Author):

The authors have improved the manuscript considerably by elaborating on the defect mediated carrier localization. They have also addressed most of my comments. However, the extrinsic contribution to the large HR factors remains largely speculative, in my opinion, while being entirely plausible. Experimentally, this could be verified either by tuning the defect density or the nature of the defects via materials engineering or time-resolving the carrier localization, both of which are probably out of the scope of the current work. Having said that, some recent time resolved X-ray measurements, which capture the lattice dynamics and the polaron formation around the defects (Nature materials 20 (5), 618-623, may be also Phys. Rev. Res. 3, 023147) may further enhance and contextualize the discussion.

Lastly, the authors claim that the perovskite lattice is harmonic at 3.8 K. This will be the case if the phonon dephasing is entirely mediated by scattering with thermally excited phonons. However, there may also be contributions from defect scattering to the phonon anharmonicity – example: Appl. Phys. Lett., 2000, 76, 1258–1260 – which will still be dominant at low temperatures. Thus, Frohlich model may after all be insufficient/incorrect for the metal halide perovskites.

Overall, the manuscript can be considered for publication after the addressing the two minor comments stated above.

Reviewer #2 (Remarks to the Author):

The authors have carefully and satisfactorily addressed the comments by the reviewer. The reviewer has no further comments and supports publication of this interesting article.

Reviewer #3 (Remarks to the Author):

I am very gratified by the authors' extensive improvements to their manuscript. I am therefore happy to recommend the article for publication in Nature Communications.

REVIEWER COMMENTS

Reviewer #1 (Remarks to the Author):

Comment 1

The authors have improved the manuscript considerably by elaborating on the defect mediated carrier localization. They have also addressed most of my comments. However, the extrinsic contribution to the large HR factors remains largely speculative, in my opinion, while being entirely plausible. Experimentally, this could be verified either by tuning the defect density or the nature of the defects via materials engineering or time-resolving the carrier localization, both of which are probably out of the scope of the current work. Having said that, some recent time resolved X-ray measurements, which capture the lattice dynamics and the polaron formation around the defects (Nature materials 20 (5), 618-623, may be also Phys. Rev. Res. 3, 023147) may further enhance and contextualize the discussion.

Response

We are glad to have received such a positive reaction to the changes we have implemented, many of which were prompted by the criticism and suggestions of Reviewer 1 in the previous round of review. We further remark that the reviewer is correct, in that the suggested additional experimental work is out of scope for this paper, since we currently address much more than the extrinsic contribution to the Huang-Rhys factor, and further investigations on this topic would undoubtedly shift the focus of the discussion at hand. That being said, we appreciate the additional references and are happy to include them into our text, as we feel they can indeed enhance our discussion and provide further context.

We note that the first reference, discussing X-ray measurements, highlights the structural modifications in the perovskite lattice upon photoexcitation. The reference employs a technique which the authors describe as ‘sensitive to the formation of point-defect-like local structural rearrangements’, meaning that it is capable of directly probing the lattice relaxation during polaron formation. As such, the reference constitutes the first direct observation of polaron formation in lead halide perovskites, and can add to our discussion by showing that polarons are inextricably involved in, and thus define, carrier behavior in lead halide perovskites. Therefore, this reference has been added to the introduction section of our manuscript. (List of Changes #1)

The second reference shows that polaronic excitons in lead halide perovskites may be strongly self-trapped at room temperature, and this effect can be attributed to the dynamic disorder of the system (i.e. the anharmonicity of the lattice). This is a sign that lattice anharmonicity may enhance exciton-phonon coupling in lead halide perovskites, and we have therefore added this reference to our bibliography (List of Changes, #2). We note however that the reference assumes a strong central peak in the vibrational response of the lattice. As already mentioned in the text, the central peak is not observed at low temperatures, neither in our experiment nor in the available literature (including, most significantly, Yaffe et al, Phys. Rev. Lett. 118, 136001 (2017), where the anharmonic effects were specifically addressed in CsPbBr₃). This is now clarified in the text, and two additional references are provided to support our claim (Guo et al, Nat. Commun. (2019) 10:1175, Green et al, J. Phys. Chem. Lett. 2015, 6, 23, 4774–4785) (List of Changes, #3). As such, the results of the second reference provided by the reviewer cannot be directly applied to our situation, and the localization model we proposed remains much more plausible. The

significance of anharmonic contributions in the context of our experiment is further discussed in our response to the second comment.

List of Changes

1.

Old version (Lines 40-41)

Here we address the large polaron model, which appropriately describes carrier-LO-phonon interactions in CsPbBr₃ and related materials.

New version (Lines 40-42)

Here we address the large polaron model, which appropriately describes carrier-LO-phonon interactions in CsPbBr₃ and related materials, as evidenced by the direct observation of the transient elastic strain fields associated with polaron formation¹⁴.

2.

Old version (Lines 284-290)

Finally, we recall from the introduction that anharmonicity effects may influence carrier-phonon scattering in lead halide perovskites¹⁴⁻¹⁷.

New version (Lines 286-289)

Finally, we recall from the introduction that anharmonicity effects may influence carrier-phonon scattering in lead halide perovskites¹⁴⁻¹⁷. In fact, the anharmonic nature of the CsPbBr₃ lattice at room temperature may even be linked to strong self-trapping of excitonic polarons⁶⁶.

3.

Old version (Lines 287-290)

Since this limit lies well above our experimental temperature of 3.8 K, we can therefore surmise that the experimentally-obtained S factor does not contain any anharmonic contributions, and thus cannot be intrinsically accounted for.

New version (Lines 289-295)

However, strong anharmonic contributions at room temperature are known to give rise to a strong central peak in the Raman spectrum of a material – an effect which has not been observed at cryogenic temperatures in lead halide perovskites, neither in previous experiments below 77 K^{14,67,68}, nor in our own experiment, presented here. We can therefore surmise that the lattice anharmonicity is significantly reduced at the low temperature of our experiment (3.8 K), and thus the S factor cannot be intrinsically accounted for.

Comment 2

Lastly, the authors claim that the perovskite lattice is harmonic at 3.8 K. This will be the case if the phonon dephasing is entirely mediated by scattering with thermally excited phonons. However, there may also be contributions from defect scattering to the phonon anharmonicity – example: Appl. Phys. Lett., 2000, 76, 1258–1260 – which will still be dominant at low temperatures. Thus, Fröhlich model may after all be insufficient/incorrect for the metal halide perovskites.

Response

We agree that phonon-defect scattering may, in general, play a role in lead halide perovskites, as proposed by the reviewer. We have thus added the suggested reference to our bibliography (List of Changes #4). Furthermore, our claims that there are no anharmonic effects at cryogenic temperatures have been toned down, in order to allow the possibility of phonon-defect scattering playing a role. (List of Changes #5 & #6)

We must however remark that, while factors such as non-adiabaticity and anharmonicity can, and sometimes should, be considered as higher-order corrections to the Fröhlich model in lead halide perovskites, the Fröhlich model still provides an excellent first-order approximation of carrier behavior, especially at low temperature. In our particular case, the Fröhlich model excellently reproduces the experimentally-recorded progression of LO phonon replicas in the multiphonon Raman spectrum of Figure 1. Thus, since the model is successful in reproducing the experimental data, it is reasonable to conclude that the initial assumptions were correct, i.e. that anharmonicity does not significantly affect scattering at low temperatures.

We also note that substantial evidence exists for the effects of anharmonicity on the vibrational dynamics of lead halide perovskites – see Debnath et al, Nat. Commun. (2021) 12:2629. This reference has also been added to our manuscript for further context (List of Changes #4). Therein, the authors argue that FAPbI₃ is strongly anharmonic at room temperature, whereas its counterpart FAPbBr₃ is closer to the harmonic approximation. Pump-probe measurements thereby revealed that the increased anharmonicity of the former produces additional second-order carrier-phonon interactions. The enhancement of multiphonon interactions in the case of strong anharmonicity should thus lead to significant deviations from the progression of phonon replicas derived under the Fröhlich model. We hereby reiterate that this is not the case in our work, as both the resonant Raman spectrum of Figure 1, as well as the low-temperature PL spectrum (see Iaru et al, ACS Nano 2017, 11, 11024-11030) could be excellently reproduced without correcting for anharmonic contributions. We thus maintain that all anharmonic contributions, including those due to phonon-defect scattering, are negligible in our experiment.

List of Changes

4.

Old version (Lines 46-50)

We note, however, that anharmonic lattice vibrations may account for discrepancies between first-principles calculations within the Fröhlich model and experimentally-measured carrier mobilities.¹⁴⁻¹⁷

New version (Lines 43-48)

We note, however, that anharmonic lattice vibrations may account for discrepancies between first-principles calculations within the Fröhlich model and experimentally-measured carrier mobilities¹⁵⁻¹⁸. Additionally, strong anharmonicity has been shown to significantly modify the vibrational dynamics of lead halide perovskites¹⁹. Finally, phonon-defect scattering can lead to phonon dephasing, which may further disrupt the fully-harmonic picture of the perovskite lattice that the Fröhlich model assumes²⁰.

5.

Old version (Lines 53-56)

A good starting point for directly probing the Fröhlich interaction, in the absence of anharmonicity effects, is to conduct measurements at cryogenic temperatures, noting that the impact of the Fröhlich interaction on carrier behavior should only increase towards higher temperatures, as LO phonons become thermally available.

New version (Lines 52-55)

A good starting point for directly probing the Fröhlich interaction is to conduct measurements at cryogenic temperatures, noting that the impact of the Fröhlich interaction on carrier behavior should only increase towards higher temperatures, as LO phonons become thermally available.

6.

Old version (Lines 287-290)

Since this limit lies well above our experimental temperature of 3.8 K, we can therefore surmise that the experimentally-obtained S factor does not contain any anharmonic contributions, and thus cannot be intrinsically accounted for.

New version (Lines 293-296)

We can therefore surmise that the lattice anharmonicity is significantly reduced at the low temperature of our experiment (3.8 K), and thus the S factor cannot be intrinsically accounted for.